# Momentum Multi-Marginal Schrödinger Bridge Matching

**Panagiotis Theodoropoulos[1], Augustinos D. Saravanos[1],**
**Evangelos A. Theodorou[1,*], Guan-Horng Liu[2,*]**
[1]Georgia Institute of Technology, [2]FAIR at Meta, [*]Equal advising

## Abstract

Understanding complex systems by inferring trajectories from sparse sample snapshots is a fundamental challenge in a wide range of domains, e.g., single-cell biology, meteorology, and economics. Despite advancements in Bridge and Flow matching frameworks, current methodologies rely on pairwise interpolation between adjacent snapshots. This hinders their ability to capture long-range temporal dependencies and potentially affects the coherence of the inferred trajectories. To address these issues, we introduce **Momentum Multi-Marginal Schrödinger Bridge Matching (3MSBM)**, a novel matching framework that learns smooth measure-valued splines for stochastic systems that satisfy multiple positional constraints. This is achieved by lifting the dynamics to phase space and generalizing stochastic bridges to be conditioned on several points, forming a multi-marginal conditional stochastic optimal control problem. The underlying dynamics are then learned by minimizing a variational objective, having fixed the path induced by the multi-marginal conditional bridge. As a matching approach, 3MSBM learns transport maps that preserve intermediate marginals throughout training, significantly improving convergence and scalability. Extensive experimentation in a series of real-world applications validates the superior performance of 3MSBM compared to existing methods in capturing complex dynamics with temporal dependencies, opening new avenues for training matching frameworks in multi-marginal settings.

## 1   Introduction

Transporting samples between probability distributions is a fundamental problem in machine learning. Diffusion Models (DMs;[Ho et al., 2020, Song et al., 2020]) constitute a prominent technique in generative modeling, which employ stochastic mappings through Stochastic Differential Equations (SDEs) to transport data samples to a tractable prior distribution, and then learn to reverse this process [Anderson, 1982, Vincent, 2011]. However, diffusion models present several limitations, e.g., the lack of optimality guarantees concerning the kinetic energy for their generated trajectories [Shi et al., 2023]. To address these shortcomings, principled approaches that stem from Optimal Transport [Villani et al., 2009] have emerged that aim to minimize the transportation energy of mapping samples between two marginals, $\pi_0$ and $\pi_1$. In this vein, the Schrödinger Bridge (SB; [Schrödinger, 1931])—equivalent to Entropic Optimal Transport (EOT;[Cuturi, 2013, Peyré et al., 2019])—has been one of the most prominent approaches used in generative modeling [Vargas et al., 2021]. This popularity has been enabled by recent remarkable advancements of matching methods [Lipman et al., 2022, Liu et al., 2022a]. Crucially, these matching-based frameworks circumvent the need to cache the full trajectories of forward and backward SDEs, mitigate the time-discretization and "forgetting" issues encountered in earlier SB techniques, and maintain a feasible transport map throughout training. This renders them highly scalable and stable methods for training the SB [Shi et al., 2023, Gushchin et al., 2024a, Peluchetti, 2023, Liu et al., 2024, Rapakoulias et al., 2024].

39th Conference on Neural Information Processing Systems (NeurIPS 2025).

Table 1: Comparison between our 3MSBM and state-of-the-art multi-marginal algorithms in 1) simulation-free training, 2) smooth and coherent trajectories and 3) globally optimal coupling.

| | Simulation-Free Training | Smooth Trajectories | Global Coupling |
|---|---|---|---|
| DMSB [Chen et al., 2023a] | ✗ | ✓ | ✓ |
| MMFM [Rohbeck et al., 2024] | ✓ | ✓ | ✗ |
| SBIRR [Shen et al., 2024] | ✓ | ✗ | ✗ |
| MMSFM [Lee et al., 2025] | ✓ | ✗ | ✗ |
| Smooth SB [Hong et al., 2025] | ✗ | ✓ | ✓ |
| **3MSBM (Ours)** | ✓ | ✓ | ✓ |

However, many complex real-world scenarios provide separate measurements at coarse time intervals [Chen et al., 2023a]. In this case, solving several distinct SB problems between adjacent marginals and connecting the ensuing bridges leads to suboptimal trajectories that fail to account for temporal dependencies or model the dynamics without discontinuities [Lavenant et al., 2024]. To address this issue, a generalization of the SB problem is considered: the multi-marginal Schrödinger Bridge (mmSB) [Chen et al., 2019], in which the dynamics are augmented to the phase space. Incorporating the velocity and coupling it with the

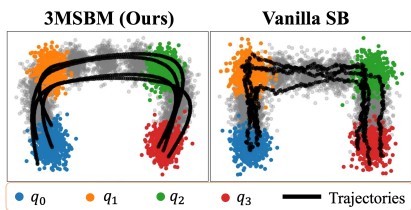

Figure 1: Trajectory comparison between by vanilla SB, and our 3MSBM.

position leads to smooth trajectories traversing multiple time-indexed marginals in position space [Dockhorn et al., 2021, Chen et al., 2023a], as illustrated in Figure 1. This approach enables us to capture the underlying dynamics and better leverage the information-rich data of complex systems such as cell dynamics [Yeo et al., 2021, Zhang et al., 2024a], meteorological evolution [Franzke et al., 2015], and economics [Kazakevičius et al., 2021]. This wide applicability of multi-marginal systems in real-world applications has spurred significant interest in developing algorithms to address these problems. However, existing methodologies either sacrifice optimality for scalability or vice versa.

On one end of the spectrum, many recent methods optimize locally between adjacent marginals, often exhibiting substantial scalability, yet failing to recover the global coupling. For instance, Shen et al. [2024] propose a simulation-free iterative scheme that solves mmSB by performing trajectory inference with pairwise bridge optimization. Nevertheless, this pairwise formulation cannot enforce global trajectory consistency and thus depends on informative priors, limiting practicality in real-world settings. Similarly, Multi-Marginal Flow Matching [MMFM; Rohbeck et al. [2024]] uses deterministic cubic-spline interpolation with precomputed piecewise couplings, impairing temporal coherence and dynamical fidelity. Furthermore, the stochastic counterpart of MMFM [Multi-Marginal Stochastic Flow Matching [(MMSFM); Lee et al. [2025]] optimizes measured-value splines over overlapping triplets of marginals, improving on MMFM's pairwise scheme; however, the lack of global coupling and the first-order dynamics still limit smooth, temporally coherent trajectories. On the other hand, methods that solve the mmSB without local approximations can recover the global coupling but generally scale poorly. Deep Momentum Multi-Marginal Schrödinger Bridge [DMSB; Chen et al. [2023a]] solves for the global mmSB coupling in phase space using Bregman iterations. However, it suffers from scalability limitations due to the need to cache full SDE trajectories, leading to computational bottlenecks, error accumulation, and potential instability. Lastly, modeling the reference dynamics as smooth Gaussian paths achieved temporally coherent and smooth trajectories [Hong et al., 2025], though the belief propagation prohibits scaling in high dimensions. Table 1 summarizes the key attributes of these approaches alongside our proposed methodology.

In this work, we introduce **Momentum Multi-Marginal Schrödinger Bridge Matching (3MSBM)**, a novel scalable matching algorithm that solves the mmSB in the phase space. We lift the dynamics to phase space and minimize path acceleration, resulting in smooth low-curvature trajectories. This preserves trends often lost in first-order models—crucial under sparse or irregular observations—and captures non-linear transitions and inflection points for more realistic interpolation. We begin by deriving momentum Brownian bridges, yielding closed-form expressions for conditional bridges that traverse any arbitrary number of marginals. This eliminates the need for costly numerical integrations, improving efficiency and avoiding error accumulation. The resulting conditional path—optimal under stochastic optimal control and satisfying all positional marginal constraints—is then held fixed while

we match a parameterized drift to the prescribed trajectory, enabling simulation-free drift learning. Iterating these steps converges to the globally optimal mmSB coupling. Algorithmically, our 3MSBM shares similar attributes with other matching frameworks [Liu et al., 2024], as the model-induced marginals remain close to the ground truth throughout the optimization, unlike prior mmSB solvers that align with the targets only at convergence. Empirical results verify the efficiency and scalability of our framework, handling high-dimensional problems and outperforming state-of-the-art methods tailored to tackle multi-marginal settings. Our contributions are summarized as follows:

- We propose 3MSBM, a novel matching algorithm for learning smooth interpolation, while preserving multiple marginal constraints.
- We present a theoretical analysis extending the concept of Brownian Bridges to second-order systems with the capacity to handle arbitrarily many marginals.
- Unlike prior mmSB methods, e.g. [Chen et al., 2023a], our method enjoys provable stable convergence and admits a map that satisfies the marginal constraints throughout training.
- Extensive experimentation demonstrates the enhanced scalability of 3MSBM, with respect to the dimensionality of the input data and the number of marginals.

## 2   Preliminaries

### 2.1   Schrödinger Bridge

The Schrödinger Bridge can be obtained through the following Stochastic Optimal Control (SOC) formulation, trying to find the unique non-linear stochastic process $x_t \in \mathbb{R}^d$ between marginals $\pi_0,\ \pi_T$ that minimizes the kinetic energy [Chen et al., 2016]

$$\min_{u_t, p_t} \int_0^T \mathbb{E}_{p_t}[\|u_t\|^2]dt \quad \text{s.t. } dx_t = u_t dt + \sigma dW_t, \qquad x_0 \sim \pi_0, \qquad x_T \sim \pi_T \tag{1}$$

resulting in the stochastic equivalent [Gentil et al., 2017] of the fluid dynamic formulation in OT [Benamou and Brenier, 2000]. Specifically, the optimal drift of Eq. (1) generates the optimal probability path $p_t$ of the dynamic Schrödinger Bridge (dSB) between the marginals $\pi_0$, and $\pi_T$. More recently, the advancement of matching algorithms [Gushchin et al., 2024b, De Bortoli et al., 2023] led to the development of highly efficient and scalable SB Matching algorithms. Representing the marginal probability path $p_t$ as a mixture of endpoint-conditioned bridges, $p_t = \int p_{t|0,T}, d\pi_{0,T}(x_0, x_T)$ [Léonard, 2013], motivates a two-step alternating training scheme [Theodoropoulos et al., 2024]. The first step entails fixing the coupling $\pi_{0,T}$ by drawing pairs of samples $(x_0, x_T)$ and optimizing intermediate bridges between the drawn pairs. Subsequently, the parameterized drift $u_t^\theta$ is matched given the prescribed marginal path from the previous step, progressively refining the coupling induced by $u_t^\theta$. At convergence, these steps aim to construct a stochastic process whose coupling matches the static solution of the SB, i.e. $\pi_{0,T}^\star$, and optimally interpolates the coupling $(x_0, x_1) \sim \pi_{0,T}^\star$.

### 2.2   Momentum Multi-Marginal Schrödinger Bridge

The Momentum Multi-Marginal Schrödinger Bridge (mmSB) extends the objective in Eq. (1) to traverse multiple marginals constraints. Additionally, the dynamics are lifted into second-order, incorporating the velocity, denoted with $v_t \in \mathbb{R}^d$, along with the position $x_t$. Consequently, the marginal distributions are also augmented $\pi_n := \pi_n(x, v)$, for $n = \{0, 1, \ldots, N\}$, as they depend on both the position $x_t$, and the velocity $v_t$. We define the joint marginal probability path $p_t(x_t, v_t)$ for $t \in [0, T]$, and the position and velocity marginals with $q_t(x) = \int p_t(x, v)dv$, and $\xi_t(v) = \int p_t(x, v)dx$ respectively. Application of the Girsanov theorem in the phase space yields the corresponding multi-marginal phase space SOC formulation [Chen et al., 2019]

$$u_t^\star = \begin{bmatrix} 0 \\ a_t^\star \end{bmatrix} \in \arg\min_{u_t} \int \mathbb{E}_{p_t}[\|a_t\|^2]dt$$

$$dm_t = Am_t dt + u_t dt + \mathbf{g}dW_t, \quad x_n \sim q_n = \int \pi_n(x, v)dv_n, \quad n = \{0, 1, \ldots, N\}, \tag{2}$$

where $m_t = [x_t, v_t]^\intercal \in \mathbb{R}^{2d}$, $A = \begin{bmatrix} 0 & 1 \\ 0 & 0 \end{bmatrix}$, and $\mathbf{g} = \begin{bmatrix} 0 & 0 \\ 0 & \sigma \end{bmatrix}$.

# 3 Momentum Multi-Marginal Schrödinger Bridge Matching

We propose **Momentum Multi-Marginal Schrödinger Bridge Matching (3MSBM)**, a novel matching framework, which incorporates the velocity into the dynamics to learn smooth measure-valued splines for stochastic systems satisfying positional marginal constraints over time. Following recent advances in matching frameworks, our algorithm solves Eq. (2), separating the problem into two components: 1) the optimization of the intermediate path, conditioned on the multi-marginal coupling, and 2) the optimization of the parameterized drift, refining the coupling.

## 3.1 Intermediate Path Optimization

Our analysis begins by formulating a multi-marginal conditional path that satisfies numerous constraints at sparse intervals. Specifically, we first derive the optimal control expression for a phase-space conditional bridge, conditioned at $\{\bar{m}_n\}$, for $n \in \{0, 1, \ldots, N\}$. Based on this expression for the optimal control, we then obtain a recursive formula for the conditional acceleration that interpolates through a set of prescribed positions $\bar{x}_n \sim q_n(x_n)$. We denote this set of fixed points as $\{\bar{x}_n\} := \{\bar{x}_0, \bar{x}_1, \ldots, \bar{x}_N\}$, and the coupling as $q(\{x_n\})$.

**Theorem 3.1** (SOC representation of Multi-Marginal Momentum Brownian Bridge (3MBB))**.** *Consider the following momentum system interpolating among multiple marginals*

$$\min_{a_t} \int_0^1 \frac{1}{2}\|a_t\|^2 dt + \sum_{n=1}^{N}(m_n - \bar{m}_n)^{\mathsf{T}} R(m_n - \bar{m}_n) \tag{3}$$

$$s.t \quad dm_t = Am_t dt + u_t dt + \mathbf{g}dW_t, \quad m_0 = \bar{m}_0 \tag{4}$$

*We define the value function as $V_t(m_t) := \frac{1}{2}m_t^T P_t^{-1} m_t + m_t^T P_t^{-1} r_t$, where $P_t$, $r_t$ are the second-and first-order approximations, respectively. This formulation admits the following optimal control expression $u_t^{\star}(m_t) = -\mathbf{g}\mathbf{g}^T P_t^{-1}(m_t + r_t)$. For the multi-marginal bridge with $\{\bar{m}_n\}$ fixed at $\{t_n\}$, for $n \in \{0, 1, \ldots, N\}$, the dynamics of $P_t$ and $r_t$ obey the following backward ODEs*

$$\begin{aligned}
\dot{P}_t &= AP_t + P_t A^T - \sigma_t \sigma_t^T, & P_n &= (P_{n^+}^{-1} + R)^{-1}, \; \text{with } P_{N^+} = 0 \\
\dot{r}_t &= -Ar_t, & r_n &= P_n(P_{n^+}^{-1} r_{n^+} - R\bar{m}_n)
\end{aligned} \tag{5}$$

*where $P_{n^+} := \lim_{t \to t_n^+} P_t$, and $r_{n^+} := \lim_{t \to t_n^+} r_t$, for $t \in \{s : s \in (t_1, t_2) \vee (t_2, t_3), \cdots\}$.*

Our 3MBB presents a natural extension of the well-established concept of the momentum Brownian Bridge [Chen and Georgiou, 2015]. For the derivation of the multi-marginal bridge, we apply the dynamic programming principle, recursively optimizing acceleration in each segment while accounting for subsequent segments via the intermediate constraints, as illustrated in Figure 2. The terms $P_{n^+}$, and $r_{n^+}$ capture the influence of the subsequent segment through the intermediate constraints $P_n$, $r_n$, which serve as terminal conditions for the corresponding ODEs of the next segment. Importantly, from the terminal conditions in Eq. (5) it is implied that $r_t$—and hence the optimal control $u_t^{\star}$—would depend on all subsequent pinned points after $t$ $\{\bar{m}_n : t_n \geq t\}$, as shown explicitly, in the acceleration formulation derived in the next proposition.

**Proposition 3.2.** *Let $R = \begin{bmatrix} \frac{1}{c} & 0 \\ 0 & c \end{bmatrix}$. At the limit when $c \to 0$, the solution of 3MBB (Th. 3.1) admits a closed-form expression on every segment; in particular, for $t \in [t_n, t_{n+1})$:*

$$a_t^{\star}(m_t|\{\bar{x}_{n+1} : t_{n+1} \geq t\}) = C_1^n(t)(x_t - \bar{x}_{n+1}) + C_2^n(t)v_t + C_3^n(t)\sum_{j=n+1}^{N}\lambda_j \bar{x}_j, \tag{6}$$

*where $\{\bar{x}_{n+1} : t_{n+1} \geq t\}$ signifies the bridge is conditioned on the set of the ensuing points, $\lambda_j$ are static coefficients and $C_1^n(t), C_2^n(t), C_3^n(t)$ are time-varying coefficients specific to each segment. The proof, the definitions for these functions, and the $\lambda_j$ coefficient values are left for Appendix B.2.*

The expression in Eq. (6) provides a recursive formula to compute the optimal conditional bridge for the segment $t \in [t_n, t_{n+1})$. Notice that the linear combination $\sum_{j=n+1}^{N}\lambda_j x_j$ captures the dependence of each segment on all next pinned points after $t$ $\{\bar{m}_{n+1} : t_{n+1} \geq t\}$.

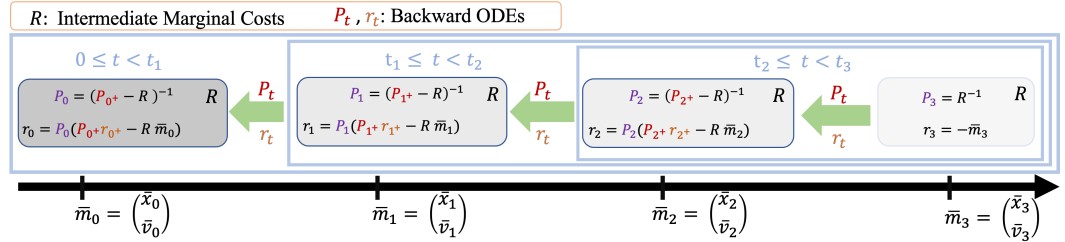

Figure 2: Visualization of the Dynamic Principle. $P_t$, $r_t$ in Eq. (5) are solved backward, propagating the influence of future pinned points in preceding segments, through the intermediate constraints.

*Remark* 3.3. Importantly, the expression for the acceleration in Eq. (6) explicitly shows that the optimal bridge does not need to converge to predefined velocities $\bar{v}_n$ at the intermediate marginals.

As $c \to 0$ in the intermediate state costs, the constraints on the joint variable $\bar{m}_n$ shift to ensure the trajectory reaches the conditioned $\bar{x}_n$ at time $t_n$, without explicitly prescribing any velocities at the intermediate points, consistent with the principles of Bridge Matching.

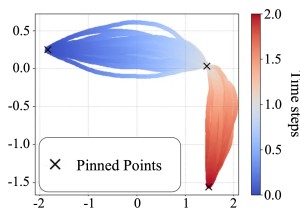

**Example - Bridge for 3 marginals** To obtain the stochastic bridge of Eq. (3) between two marginals, i.e., $N = 2$, we consider the same value function approximation as in Th. 3.1, admitting the same optimal control law $u_t^\star(m_t) = -\mathbf{g}\mathbf{g}^\intercal P_t^{-1}(m_t + r_t)$. For simplicity, let us assume $t_0 = 0$, $t_1 = 1$, and $t_2 = 2$. From Prop. 3.2, we obtain the following solution for conditional acceleration.

Figure 3: Bridge materializations among 3 pinned points.

$$\begin{cases} a^\star(m_t|, \bar{x}_2) & = \frac{3}{(t-2)^2}(\bar{x}_2 - x_t) - \frac{3}{2-t}v_t, & t \in [1, 2) \\ a^\star(m_t|\bar{x}_1, \bar{x}_2) & = \frac{30-18t}{(t-1)^2(3t-7)}(x_t - \bar{x}_1) + \frac{12(t-2)}{(t-1)(3t-7)}v_t + \frac{6}{(3t-7)}(\bar{x}_2 - \bar{x}_1), & t \in [0, 1) \end{cases} \quad (7)$$

Notice that in the segment $t \in [0, 1)$, the $\lambda$ coefficients of the linear combination $\sum_{n=1}^{2} \bar{x}_j$ in Eq. (6) are found to be: $\lambda_1 = -1, \lambda_2 = 1$, whereas for $t \in [1, 2]$ the sole coefficient is $\lambda = 0$. Appendix B.2 presents more examples of conditional accelerations with more marginals, demonstrating the dependency of the functions $C_1^n(t), C_2^n(t), C_3^n(t)$, along with the $\lambda_j$ coefficient values on the number of the following marginals. Figure 3 depicts different materializations between 3 pinned points, illustrating the convergence of the bridges to the conditioned points.

## 3.2 Bridge Matching for Momentum Systems

Subsequently, following the optimization of the 3MBB, we match the parameterized acceleration $a_t^\theta$, given the prescribed conditional probability path $p(m_t|\{\bar{x}_n : t_n \geq t\})$, induced by the acceleration $a^\star(m_t|\{\bar{x}_n : t_n \geq t\})$. Given that the 3MBB is solved for each set of points $\{x_n\}$, we can marginalize to construct the marginal path $p_t$.

**Proposition 3.4.** *Let us define the marginal path $p_t$ as a mixture of bridges $p_t(m_t) = \int p_{t|\{\bar{x}_n\}}(m_t|\{\bar{x}_n : t_n \geq t\})dq(\{x_n\})$, where $p_{t|\{\bar{x}_n\}}(m_t|\{\bar{x}_n : t_n \geq t\})$ is the conditional probability path associated with the solution of the 3MBB path in Eq 6. The parameterized acceleration that satisfies the FPE prescribed by the $p_t$ is given by*

$$a_t(t, m_t) = \frac{1}{p_t} \int a_{t|\{\bar{x}_n\}} p_{t|\{\bar{x}_n\}}(m_t|\{\bar{x}_n : t_n \geq t\})dq(\{x_n\}) \quad (8)$$

*This suggests that the minimization of the variational gap to match $a_t^\theta$ given $p_t$ is given by*

$$\min_\theta \mathbb{E}_{q(\{x_n\})}\mathbb{E}_{p_{t|\{\bar{x}_n\}}}\Big[\int_0^1 \|a_{t|\{\bar{x}_n\}} - a_t^\theta\|^2 dt\Big] \quad (9)$$

Matching the parameterized drift given the prescribed path $p_t$ leads to a more refined coupling $q(\{x_n\})$, which will be used for the conditional path optimization in the next iteration. The linearity of the system implies that we can efficiently sample $m_t = [x_t, v_t]$, $\forall t \in [0, T]$, from the conditional

**Algorithm 1** Momentum Multi-Marginal Schrödinger Bridge Matching (**3MSBM**)

1: **Input**: Marginals $q(x_0), q(x_1), \ldots, q(x_N), R, \sigma, K, T$
2: Initialize $a_t^\theta, q(\{x_n\}) := q(x_0) \otimes q(x_1) \otimes \cdots \otimes q(x_N)$, and $v_0 \sim \mathcal{N}(0, I)$
3: **repeat**
4:     **for** $j = 0$ **to** $J$ **do**
5:         Calculate $a_{t|\{\bar{x}_n\}}$ using Eq. (6) for $t$ from 0 to $T$
6:         $v_N \leftarrow \text{sdeint}(x_0, v_0, a_{t|\{x_n\}]}, \sigma, K, T)$
7:         Calculate $a_{t|\{\bar{x}_n\}}$ using Eq. (6) for $t$ from $T$ to 0
8:         $v_0 \leftarrow \text{sdeint}(x_N, v_N, a_{t|\{\bar{x}_n\}}, \sigma, K, T)$
9:     **end for**
10:    Update $a_t^\theta$, from Eq. (9) using $a_{t|\{\bar{x}_n\}}$
11:    Sample new $x_0, x_1, \ldots, x_N$ from $a_t^\theta$
12: **until** converges

---

probability path $p_{t|\{\bar{x}_n\}} = \mathcal{N}(\mu_t, \Sigma_t)$, as the mean vector $\mu_t$ and the covariance matrix $\Sigma_t$ have analytic solutions [Särkkä and Solin, 2019]. More explicitly, we can construct $m_t$ through

$$
m_t = \begin{bmatrix} X_t \\ V_t \end{bmatrix} = \begin{bmatrix} \mu_t^x \\ \mu_t^v \end{bmatrix} + \begin{bmatrix} L_t^{xx}\epsilon_0 \\ L_t^{xv}\epsilon_0 + L_t^{vv}\epsilon_1 \end{bmatrix}, \quad \text{where } \epsilon_0, \epsilon_1 \sim \mathcal{N}(0, \mathbf{I}_d) \text{ and } L_t = \begin{bmatrix} L_t^{xx} & L_t^{xv} \\ L_t^{xv} & L_t^{vv} \end{bmatrix}
\tag{10}
$$

where $L_t$ is computed following the Cholesky decomposition of the covariance matrix. The expressions for the mean vector and the covariance matrix are in Appendix C. We conclude this section on a theoretical analysis of our methodology converging to the unique multi-marginal Schrödinger Bridge solution by iteratively optimizing Eq. (3), and Eq. (9).

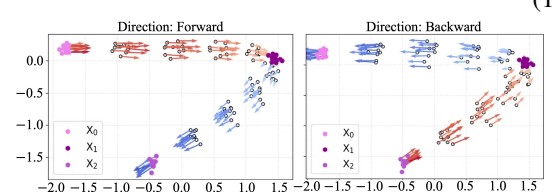

Figure 4: Iterative propagation of dynamics using the optimal conditional acceleration to approximate the velocity profile $\pi_n(v_n|x_n)$ at the intermediate marginals.

**Convergence**  We establish the convergence of our alternating scheme — between interpolating path optimization and the coupling refinement — to the unique mmSB solution. Let us denote with $\mathbb{P}$ the path measure associated with the learned dynamics, and with $\mathbb{P}^i$ the path measure of the learned dynamics at the $i^{\text{th}}$ iteration of our algorithm.

**Theorem 3.5.** *Under mild assumptions, our iterative scheme admits a fixed point solution $\mathbb{P}^\star$, i.e., $\text{KL}(\mathbb{P}^i|\mathbb{P}^\star) \to 0$, and in particular, this fixed point coincides with the unique $\mathbb{P}^{mmSB}$.*

We provide an alternative claim to the convergence proof in [Shi et al., 2023]. We establish convergence to the global minimizer, based on optimal control principles and the monotonicity of the KL divergence, expressing Eq. (2) as a KL divergence, due to the Girsanov Theorem [Chen et al., 2019].

### 3.3 Training Scheme

A summary of our training procedure is presented in Alg. 1. The first step of our alternating matching algorithm is to compute the conditional multi-marginal bridge, fixing the parameterized coupling $q^\theta(\{x_n\})$ and sampling collections of points $\{\bar{x}_n\} \sim q^\theta(\{x_n\})$. Furthermore, the initial velocity distribution $v_0 \sim \xi_0(v) = \int \pi_0(x, v)dx$ is needed, which is unknown in practice for most applications. To address this, following [Chen et al., 2023a], we initialize the velocity $v_0 \sim \mathcal{N}(0, I)$ and iteratively propagate the dynamics with the conditional acceleration in Eq. (6) for the given samples (*lines 4-9* in Alg. 1). This approximates the true conditional distribution $\pi_n(v_n|x_n)$ via Langevin-like dynamics. Notably, this process requires only a few iterations (at most $\sim 10$ iterations in our experiments), and does not involve backpropagation, thus adding negligible computational cost even in high dimensions. This iterative propagation yields the optimal conditional acceleration in Eq. (6), which induces the optimal conditional path. Solving the 3MBB for every set of $\{\bar{x}_n\}$ enables us to marginalize and construct the marginal path $p_t$. Subsequently, we fix the marginal path and match the parameterized acceleration $a_t^\theta(t, m_t)$ minimizing the variational objective in Eq. (9). This optimization induces a refined joint distribution $q^\theta(\{x_n\})$, by propagating the dynamics through the augmented SDE in Eq. (2), which will be used in the first step of the next training iteration.

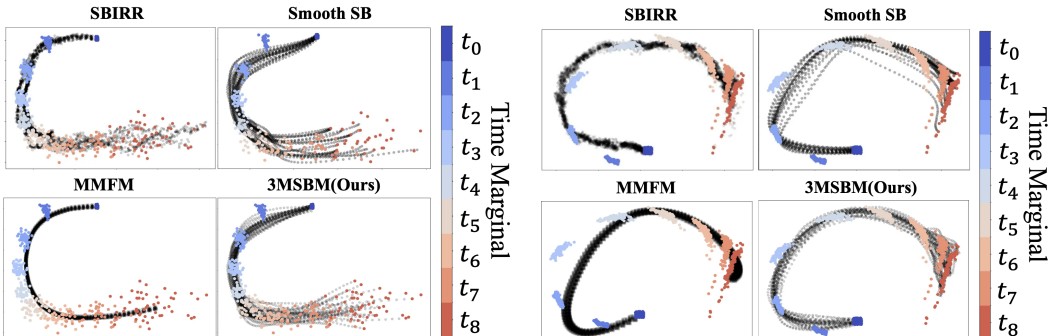

Figure 5: Trajectory comparison on LV among SBIRR, MMFM, Smooth SB, and our 3MSBM.

Figure 6: Trajectory comparison on GoM among SBIRR, MMFM, Smooth SB, and our 3MSBM.

Table 2: Mean and Standard Deviation SWD distances over 5 seeds at the left-out marginals, and average SWD from the rest points belonging to the training set on the LV data for MIOFlow, SBIRR, MMFM, Smooth SB, and 3MSBM (Lower is better).

| Method | SWD $t_1$ | SWD $t_3$ | SWD $t_5$ | SWD $t_7$ | Rest SWD |
|---|---|---|---|---|---|
| MIOFlow | 1.53±0.13 | 1.49±0.09 | 1.22±0.07 | 1.46±0.06 | 0.93±0.04 |
| SBIRR | **0.17**±0.03 | 0.16±0.03 | 0.24±0.03 | 0.48±0.05 | **0.18**±0.02 |
| MMFM | 0.21±0.02 | 0.25±0.02 | 0.39±0.03 | 0.57±0.05 | 0.22±0.01 |
| DMSB | 0.64±0.06 | 0.67±0.06 | 0.98±0.07 | 0.63±0.04 | 0.43±0.04 |
| Smooth SB | 0.29±0.03 | 0.18±0.02 | 0.11±0.02 | 0.37±0.03 | 0.23±0.02 |
| 3MSBM | 0.23±0.02 | **0.13**±0.01 | **0.15**±0.02 | **0.36**±0.03 | **0.18**±0.02 |

## 4 Experiments

We empirically validate the computational and performance benefits of our method. The simulation-free training scheme of our algorithm suggests that we avoid costly numerical simulations and approximation errors, consistent with benefits observed by prior matching methods [Shi et al., 2023, Liu et al., 2024]. For this reason, our algorithm is capable of preserving high scalability while maintaining accuracy, as demonstrated in the experiments below. We evaluate the performance of 3MSBM on synthetic and real-world trajectory inference tasks, such as Lotka-Volterra (Sec. 4.1), ocean current in the Gulf of Mexico (Sec. 4.2), single-cell sequencing (Sec. 4.4), and the Beijing air quality data (Sec. 4.3). We compare against state-of-the-art methods explicitly designed to incorporate multi-marginal settings, such Deep Momentum Multi-Marginal Schrödinger Bridge (DMSB; Chen et al. [2023a]), Schrodinger Bridge with Iterative Reference Refinement (SBIRR; Shen et al. [2024]), smooth Schrodinger Bridges (smoothSB; Hong et al. [2025]), and Multi-Marginal Flow Matching (MMFM; Rohbeck et al. [2024]), and against one additional Neural ODE-based method: MIOFlow [Huguet et al., 2022], using the same metric in all datasets: the Sliced Wasserstein Distance (SWD). Additional results for these tasks—with more metrics and baselines—are provided in Appendix E.

### 4.1 Lotka-Volterra

We first consider a synthetic dataset generated by the Lotka–Volterra (LV) equations [Goel et al., 1971], which model predator-prey interactions through coupled nonlinear dynamics. The generated dataset consists of 9 marginals in total; the even-numbered indices are used to train the model (i.e., $t_0, t_2, t_4, t_6, t_8$), and the remainder of the time points are used to assess the efficacy of our model to impute and infer the missing time points. In this experiment, we benchmarked 3MSBM against MIOFlow, SBIRR, MMFM, DMSB, and Smooth SB. Performance was evaluated using the SWD distance to the validation marginals to measure imputation accuracy, and the SWD distance to the training marginals to assess how well each method preserved the observed data during generation. The results in Table 2 and Figure 5 show that 3MSBM outperforms the baseline models in inferring the marginals at the missing points, yielding the lowest deviation from most left-out marginals, while also generating trajectories which preserve the training marginals more faithfully, as shown by the lower average SWD distance over the remaining points belonging to the training set. Additional results—with more metrics and baselines—are provided in Appendix E.2.

Table 3: Mean and SD over 5 seeds for SWD distances on the GoM for MIOFlow, SBIRR, MMFM, DMSB, Smooth SB, and 3MSBM (Lower is better).

| Method | SWD $t_1$ | SWD $t_3$ | SWD $t_5$ | SWD $t_7$ | Rest SWD |
|---|---|---|---|---|---|
| MIOFlow | 0.83 $\pm 0.06$ | 0.34 $\pm 0.03$ | 1.23 $\pm 0.09$ | 0.96 $\pm 0.06$ | 0.19 $\pm 0.03$ |
| SBIRR | 0.15 $\pm 0.03$ | **0.11** $\pm 0.02$ | 0.11 $\pm 0.03$ | 0.09 $\pm 0.04$ | 0.13 $\pm 0.04$ |
| MMFM | 0.23 $\pm 0.04$ | 0.25 $\pm 0.08$ | 0.10 $\pm 0.03$ | 0.19 $\pm 0.04$ | 0.14 $\pm 0.03$ |
| DMSB | 0.22 $\pm 0.02$ | 0.54 $\pm 0.04$ | 0.39 $\pm 0.02$ | 0.28 $\pm 0.04$ | 0.09 $\pm 0.03$ |
| Smooth SB | 0.17 $\pm 0.02$ | 0.14 $\pm 0.04$ | 0.10 $\pm 0.02$ | 0.13 $\pm 0.02$ | 0.08 $\pm 0.01$ |
| 3MSBM | **0.14** $\pm 0.02$ | 0.14 $\pm 0.02$ | **0.08** $\pm 0.01$ | **0.06** $\pm 0.01$ | **0.05** $\pm 0.01$ |

Table 4: Mean and SD over 5 seeds for SWD distances on the Beijing air quality data for MMFM, DMSB, and 3MSBM (Lower is better).

| Method | SWD $t_2$ | SWD $t_5$ | SWD $t_8$ | SWD $t_{11}$ | Rest SWD |
|---|---|---|---|---|---|
| MMFM | **17.51** $\pm 2.41$ | 23.94 $\pm 1.97$ | 32.56 $\pm 2.96$ | 39.98 $\pm 3.59$ | 30.25 $\pm 2.17$ |
| DMSB | 21.10 $\pm 2.65$ | 21.92 $\pm 1.78$ | 35.53 $\pm 3.82$ | 35.75 $\pm 4.13$ | 33.42$\pm$ 2.42 |
| 3MSBM(ours) | 17.70 $\pm 1.93$ | **9.78** $\pm 1.58$ | **22.23** $\pm$ 3.64 | **32.23** $\pm 3.76$ | **21.25** $\pm 1.63$ |

Table 5: Mean and SD over 5 seeds for the MMD and SWD on Embryoid Body (EB) dataset for SBIRR, MMFM, DMSB, and 3MSBM (Lower is better).

| Method | MMD $t_1$ | SWD $t_1$ | MMD $t_3$ | SWD $t_3$ | Rest MMD | Rest SWD |
|---|---|---|---|---|---|---|
| SBIRR | 0.71$\pm 0.08$ | 0.80$\pm 0.06$ | 0.73$\pm 0.06$ | 0.91$\pm 0.05$ | 0.47$\pm 0.05$ | 0.66$\pm 0.07$ |
| MMFM | 0.37$\pm 0.02$ | 0.59$\pm 0.04$ | *0.35*$\pm 0.04$ | 0.76$\pm 0.04$ | 0.22$\pm 0.02$ | 0.52$\pm 0.07$ |
| DMSB | 0.38$\pm 0.04$ | 0.58$\pm 0.06$ | 0.36$\pm 0.07$ | 0.54$\pm 0.06$ | 0.14$\pm 0.03$ | 0.45$\pm 0.04$ |
| 3MSBM(ours) | **0.18**$\pm 0.01$ | **0.48**$\pm 0.04$ | **0.14**$\pm 0.04$ | **0.38**$\pm 0.03$ | **0.11**$\pm 0.02$ | **0.36**$\pm 0.05$ |

## 4.2 Gulf of Mexico

Subsequently, we evaluate the efficacy of our model to infer the missing time points in a real-world multi-marginal dataset. The dataset contains ocean-current snapshots of the velocity field around a vortex in the Gulf of Mexico (GoM). It includes a total of 9 marginals, with even-indexed time points (i.e., $t_0, t_2, t_4, t_6, t_8$) used for training, and the remaining are left out to evaluate the model's ability to impute and infer missing temporal states. For this experiment, we compared our 3MSBM against MIOFlow, SBIRR, MMFM, DMSB, and Smooth SB. The metrics used to evaluate performance were: SWD distance from the left-out points, and the mean of SWD distance from the training points, capturing how well the generated trajectories of each algorithm preserve the marginals that comprised the training set. Figure 6 shows that 3MSBM generates smoother trajectories with more accurate recovery of the left-out marginals. In comparison, SBIRR produces noisier, kinked trajectories; MMFM struggles to capture the dynamics, leading to larger deviations from the left-out marginals, while Smooth SB achieves the smoothest trajectories among the baselines. These observations are further confirmed by Table 3, where 3MSBM achieves the lowest SWD distances for most validation points and better preserves the training marginals. Additional results are provided in Appendix E.3.

## 4.3 Beijing Air Quality

To further study the capacity of 3MSBM to effectively infer missing values, we also tested it in the Beijing multi-site air quality data set [Chen, 2017]. This dataset consists of hourly air pollutant data from 12 air-quality monitoring sites across Beijing. We focus on PM2.5 data, an indicator monitoring the density of particles smaller than 2.5 micrometers, between January 2013 and January 2015, across 12 monitoring sites. We employed a slightly different setup than Rohbeck et al. [2024]. We focused on a single monitoring site and aggregated the measurements collected within the same month. To introduce temporal separation between observations, we selected measurements from every other month, resulting in 13 temporal snapshots. For the imputation task, we omitted the data at $t_2$, $t_5$, $t_8$, and $t_{11}$, while the remaining snapshots formed the training set. We benchmarked our 3MSBM method against MMFM with cubic splines and DMSB. Table 4 shows that 3MSBM achieved overall better imputation accuracy, yielding the smallest Sliced Wasserstein Distance (SWD) distances, while also better preserving the marginals consisting of the training snapshots compared to the baselines. Additional details and results on more metrics are left for Appendix E.4.

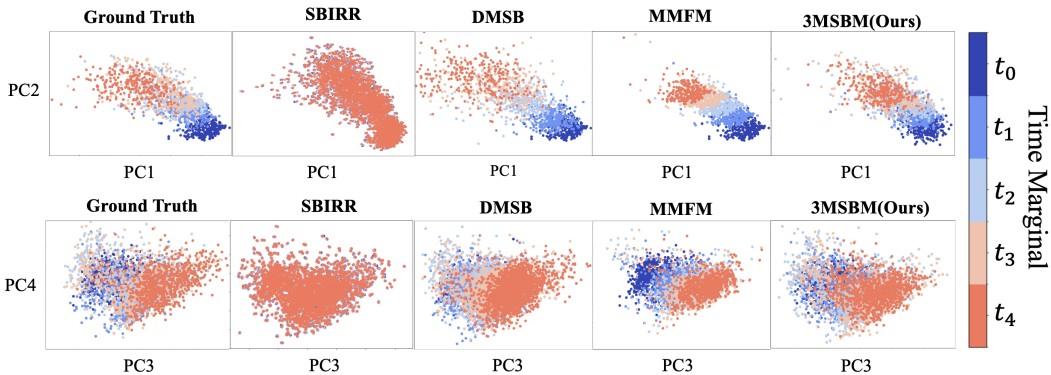

Figure 7: Comparison of the evolution of the EB dataset on the 100-dimensional PCA feature space among SBIRR, MMFM, DMSB, and 3MSBM.

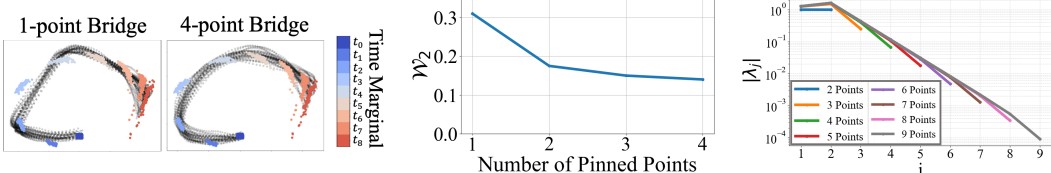

Figure 8: Conditional bridges on GoM currents using LEFT: 1 pinned point, RIGHT: 4 pinned points.

Figure 9: Mean $\mathcal{W}_2$ distance of left-out marginals for varying number of pinned points.

Figure 10: Exponential Decay of $|\lambda_j|$ coefficients for increasing number pinned points.

## 4.4 Single cell sequencing

Lastly, we demonstrate the efficacy of 3MSBM to infer trajectories in high-dimensional spaces. In particular, we use the Embryoid Body (EB) stem cell differentiation tracking dataset, which tracks the cells through 5 stages over a 27-day period. Cell snapshots are collected at five discrete day-intervals: $t_0 \in [0, 3], \quad t_1 \in [6, 9], \quad t_2 \in [12, 15], \quad t_3 \in [18, 21], \quad t_4 \in [24, 27]$. The training set consists of the even-indexed time-steps (i.e., $t_0, t_2, t_4$), while the rest are used as the validation set. We used the preprocessed dataset provided by [Tong et al., 2020, Moon et al., 2019], embedded in a 100-dimensional principal component analysis (PCA) feature space. We compare 3MSBM with SBIRR, MMFM, and DMSB, evaluating performance using Sliced Wasserstein Distance (SWD)—as in prior tasks—and additionally Maximum Mean Discrepancy (MMD). As in previous experiments, we assessed the quality of imputed marginals and the preservation of training marginals. As shown in Table 5, 3MSBM consistently outperforms existing methods across all metrics, achieving significantly more accurate imputation of missing time points and recovering population dynamics that closely match the ground truth, as illustrated in Figure 7. While DMSB also generated accurate PCA reconstructions (Figure 7), it is noted that it required approximately 2.5 times more training time than 3MSBM. A detailed comparison of the resource requirements differences between DMSB and 3MSBM is given in Sec. 4.6. Further single-cell sequencing setup and expanded results, with additional baselines and more metrics, are deferred to App. E.5.

## 4.5 Ablation study on number of pinned points

We ablate the number of pinned points used in Eq. (6), by modifying the linear combination as follows: $\sum_{j=n}^{K} \lambda_j \bar{x}_j$, starting from $K = n + 1$, namely including only the nearest fixed point, and incrementally adding more up to $K = N$. Figure 14 demonstrates that incorporating multiple pinned points significantly improves performance on the GoM dataset compared to using only the next point, enabling better inference of the underlying dynamics, as illustrated in Figure 8, albeit these benefits plateau beyond $K = n + 3$. This phenomenon is further explained by the exponential decay of coefficients $|\lambda_j|$ for increasing number, depicted in Figure 10, thus rendering distant points negligible. Notice that for the bridge using the next 2 pinned points, it holds $|\lambda_1| = |\lambda_2| = 1$ as found in Section 3.1. Consequently, practical implementations can adopt a truncated conditional policy, considering only the next $k$ pinned points, thereby significantly improving efficiency without sacrificing accuracy.

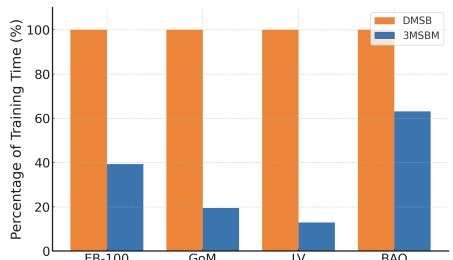

Figure 11: Training time percentage (%) comparison between 3MSBM and DMSB

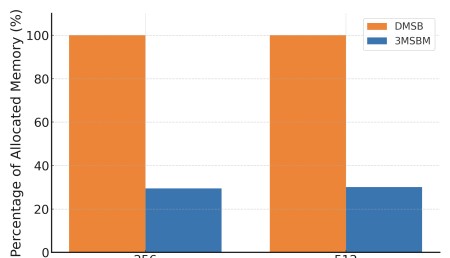

Figure 12: Allocated memory percentage (%) comparison between 3MSBM and DMSB

Table 6: Per-epoch training time (sec) increasing the marginals $N$, and the percent change [%].

| Num. of marginals | 3 | 4 | 5 |
|---|---|---|---|
| Time [s] | 773 | 789 | 812 |
| Percent Increase [%] | – | 2.1 | 5.1 |

Table 7: Percentage of time spent in 3MBB and the Bridge Matching.

| Dataset | Cond. Bridge | Bridge Matching |
|---|---|---|
| EB-100 | 3.3% | 96.7% |
| GoM | 7.4% | 92.6% |

### 4.6 Scalability of 3MSBM

We empirically demonstrate the scalability of 3MSBM. Notably, the matching-based training with the closed-form conditional bridge from 3MBB (Eq. 6)—obviating numerical integration—removes both computational overhead and approximation error.

**Computational Resources** Since 3MSBM and DMSB address the same problem, i.e the mmSB problem in Eq. 2, through a different training scheme (matching-based and IPF-based respectively), we report the resources needed by each method. In particular, Figures 11 and 12 demonstrate that our 3MSBM is faster in wall-clock time on every dataset, while also requiring significantly less memory —easily handling the high-dimensional single-cell sequencing task.

**Ablation on the number of marginals** Next, the number of marginals on EB-100 and GoM is ablated, i.e. increasing the total number of marginals $N$, while holding all other hyperparameters fixed (e.g., NFE, batch size, model size), and report per-epoch training time. In Table 6, it is shown that varying $N$ has a negligible impact on the per-epoch training time, indicating that 3MSBM scales well with the number of marginals. This insensitivity stems from our algorithmic design, as the analytic form of our conditional-bridge step is independent of $N$, and crucially.

**Comparison between 3MBB and Matching** Finally, we present a breakdown of the time percentage attributed in each of the two steps of our algorithm: i) the iterative propagation of the conditional dynamics *(lines 4-9 in Alg. 1)* and ii) the Bridge Matching step *(line 10 in Alg. 1)* in the EB-100 task. Table 7 demonstrates that the computational complexity introduced by the iterative propagation of the conditional dynamics, remains negligible compared to the Bridge Matching step.

## 5 Conclusion and Limitations

In this work, we developed 3MSBM, a novel matching algorithm that infers temporally coherent trajectories from multi-snapshot datasets, showing strong performance in high-dimensional settings and many marginals. Our work paves new ways for learning dynamic processes from sparse temporal observations, addressing a universal challenge across various disciplines. For instance, in single-cell biology, where we can only have access to snapshots of data, our 3MSBM offers a principled way to reconstruct unobserved trajectories, enabling insights into gene regulation, differentiation, and drug responses. While 3MSBM significantly improves scalability over existing multi-marginal methods, certain limitations remain. In particular, its effectiveness in densely sampled high-dimensional image spaces, such as those encountered in video interpolation, remains unexplored. As future work, we aim to extend the method to capture long-term dependencies in large-scale image settings.

## Acknowledgements

The authors would like to thank Tianrong Chen for the fruitful discussions and insightful comments that helped shape the direction of this work. This research is partially supported by the DARPA AIQ program through the DARPA CMO contract number HR00112520010. We would like to thank Dr. Pat Shafto, AIQ Program Manager, for useful technical discussions. Augustinos Saravanos acknowledges financial support by the A. Onassis Foundation Scholarship.

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

# Contents

## A  Summary of Notation

In the Table below, we summarize the notation used throughout our work.

Table 8: Notation

| | |
|---|---|
| $t$ | Time coordinate |
| $x_t$ | Position |
| $v_t$ | Velocity |
| $a_t$ | Acceleration |
| $m_t$ | Augmented Variable |
| $R$ | Soft marginal constraint |
| $q_n(x_n)$ | $n^{\text{th}}$ Positional marginal |
| $\xi_n(v_n)$ | $n^{\text{th}}$ Velocity marginal |
| $\pi_n(x_n, v_n)$ | $n^{\text{th}}$ Augmented marginal |
| $p_t(x_t, v_t)$ | Augmented probability path |
| $\lambda^{(n)}$ | Coefficients of impact of future pinned points of the $n^{\text{th}}$ segment |
| $C^{(n)}(t)$ | Functions shaping the cond. bridge of the $n^{\text{th}}$ segment |
| $\{x_n\}$ | Coupling over $n$ marginals |
| $V_t(m_t)$ | Value Function |
| $P_t, r_t$ | Value function second and first order approximations |
| $\mu_t$ | Gaussian path mean vector |
| $\Sigma_t$ | Gaussian path covariance matrix |
| $L_t$ | Cholesky decomposition on the cov. matrix |

# B  Proofs

## B.1  Proof of Theorem 3.1

**Theorem B.1** (SOC representation of Multi-Marginal Momentum Brownian Bridge (3MBB)). *Consider the following momentum system interpolating among multiple marginals*

$$u_t^\star = \begin{bmatrix} 0 \\ a_t^\star \end{bmatrix} \in \arg\min_{a_t} \int_0^1 \frac{1}{2}\|a_t\|^2 dt + \sum_{n=1}^N (m_n - \bar{m}_n)^{\mathsf{T}} R(m_n - \bar{m}_n) \tag{11}$$

$$s.t \quad dm_t = Am_t dt + u_t + \mathbf{g}dW_t, \quad m_0 = \bar{m}_0. \tag{12}$$

*We define the value function as $V_t(m_t) := \frac{1}{2}m_t^T P_t^{-1} m_t + m_t^T P_t^{-1} r_t$, where $P_t$, $r_t$ are the second- and first-order approximations, respectively. This formulation admits the following optimal control expression $u_t^\star(m_t) = -\mathbf{g}\mathbf{g}^T P_t^{-1}(m_t + r_t)$. For the multi-marginal bridge with $\{\bar{m}_n\}$ fixed at $\{t_n\}$, for $n \in \{0, 1, \ldots, N\}$, the dynamics of $P_t$ and $r_t$ obey the following backward ODEs*

$$\dot{P}_t = AP_t + P_t A^T - \sigma_t \sigma_t^T, \qquad P_n = (P_{n+}^{-1} + R)^{-1}, \text{ with } P_{N+} = 0 \tag{13}$$

$$\dot{r}_t = -Ar_t, \qquad r_n = P_n(P_{n+}^{-1} r_{n+} - R\bar{m}_n) \tag{14}$$

*where $P_{n+} := \lim_{t \to t_n^+} P_t$, and $r_{n+} := \lim_{t \to t_n^+} r_t$, for $t \in \{s : s \in (t_1, t_2) \vee (t_2, t_3), \cdots\}$.*

*Proof.* We start our analysis by considering the second-order approximation of the value function:

$$V(t, m_t) = \frac{1}{2}m_t^{\mathsf{T}} Q_t m_t + r_t^{\mathsf{T}} Q_t m_t \tag{15}$$

where $Q_t$ and $r_t$ serve as second and first-order approximations. From the Bellman principle and application of the Ito's Lemma to the value function, we obtain the Hamilton-Jacobi-Bellman (HJB) Equation:

$$V_t + \min_u \left[ \frac{1}{2}\mathbb{E}\big[\|u_t\|^2 dt\big] + V_m^{\mathsf{T}}(Am_t + u_t) \right] + \frac{1}{2}Tr(V_{mm}\mathbf{g}\mathbf{g}^{\mathsf{T}}) = 0 \tag{16}$$

Solving for the optimal control $u_t^\star$, we obtain:

$$u_t^\star = -\mathbf{g}\mathbf{g}^{\mathsf{T}} V_m = -\mathbf{g}\mathbf{g}^{\mathsf{T}} Q(m_t + r_t) \tag{17}$$

Thus plugging it back to the HJB, we can rewrite it

$$V_t - \frac{1}{2}V_m \mathbf{g}\mathbf{g}^{\mathsf{T}} V_M + V_m^{\mathsf{T}} Am + \frac{1}{2}Tr(V_{mm}\mathbf{g}\mathbf{g}^{\mathsf{T}}) = 0 \tag{18}$$

Recall the definition for the value function

$$V(t, m_t) = \frac{1}{2}m_t^{\mathsf{T}} Qm_t + r_t Qm_t$$

Substituting for the definition of the value function in the HJB yields the following PDE

$$\frac{1}{2}m_t^{\mathsf{T}}\dot{Q}m_t + \dot{r}^{\mathsf{T}} Qm_t + r_t^{\mathsf{T}}\dot{Q}m_t - \frac{1}{2}m_t^{\mathsf{T}} Q\mathbf{g}\mathbf{g}^{\mathsf{T}} Qm_t - r_t^{\mathsf{T}} Q\mathbf{g}\mathbf{g}^{\mathsf{T}} Qm_t + m_t^{\mathsf{T}} QAm_t + r_t^{\mathsf{T}} QAm_t + \frac{1}{2}Tr(V_{mm}\mathbf{g}\mathbf{g}^{\mathsf{T}}) = 0 \tag{19}$$

Grouping the terms of the PDE quadratic in $m_t$, we obtain the Riccati Equation for $Q_t$

$$-\dot{Q} = A^{\mathsf{T}} Q + QA - Q\mathbf{g}\mathbf{g}^{\mathsf{T}} Q \tag{20}$$

Then, grouping the linear terms yields

$$\dot{r}_t^{\mathsf{T}} Q + r_t^{\mathsf{T}}\dot{Q} - r_t^{\mathsf{T}} Q\mathbf{g}\mathbf{g}^{\mathsf{T}} Q + r_t^{\mathsf{T}} QA = 0 \tag{21}$$

Now, notice that substituting the Ricatti in Eq. 20 into Eq. 21, one obtains

$$\dot{r}_t = -Ar_t \tag{22}$$

The solution of this ODE is

$$r_t = \Phi(t, s)r_s \tag{23}$$

where $\Phi(t, s)$ is the state transition function of the following dynamics $dm_t = A(t)m_t dt$, from $t$ to $s$ and is defined as $\Phi(t, s) = \begin{bmatrix} 1 & t-s \\ 0 & 1 \end{bmatrix}$. Finally, we define $P(t) := Q(t)^{-1}$, and modify the Ricatti in Eq. 20 as follows

$$-\dot{Q} = A^\intercal Q + Q^{-1}QAQ^{-1} - Q\mathbf{g}\mathbf{g}^\intercal Q$$
$$-Q^{-1}\dot{Q}Q^{-1} = Q^{-1}A^\intercal QQ^{-1} + Q^{-1}QAQ^{-1} - Q^{-1}Q\mathbf{g}\mathbf{g}^\intercal QQ^{-1} \tag{24}$$
$$\dot{P} = AP + PA^\intercal - \mathbf{g}\mathbf{g}^\intercal$$

yielding the Lyapunov equation

$$\dot{P} = AP + PA^\intercal - \mathbf{g}\mathbf{g}^\intercal \tag{25}$$

Therefore, we have proved the desired ODEs for the first and second-order approximations $r_t, P_t$ of the value function. Now, we have to determine the expressions for the terminal conditions for the ODEs in each segment.

Note that it follows from the dynamic principle that these ODEs are backward, therefore ensuing segments will affect previous. These terminal conditions carry the information from the subsequent segments. More specifically, assume a multi-marginal process with $\{\bar{m}_{n+1}\}$ pinned at $\{t_{n+1}\}$. Now let us consider the two segments from both sides:

- Segment $n$: for $t \in [t_n, t_{n+1}]$
- Segment $n + 1$: for $t \in [t_{n+1}, t_{n+2}]$

To solve $P_t, r_t$ for Segment $n$, we have to account for the effect of Segment $n + 1$. To compute the value function at $t_{n+1}$, accounting for the impact from Segment $n + 1$

$$V_{n+1} = V_{n+1^+} + \frac{1}{2}(m_{n+1} - \bar{m}_{n+1})^T R_{n+1}(m_{n+1} - \bar{m}_{n+1}) \tag{26}$$

where the first term encapsulates the impact from Segment $n + 1$. More explicitly, for the Segment $n + 1$, at $t = t_{n+1}$, we define the value function as

$$\begin{aligned} V_{n+1^+} &= \frac{1}{2}m_{n+1^+}^T Q_{n+1^+}m_{n+1^+} + r_{n+1^+}Q_{n+1^+}m_{n+2^+} \\ &= \frac{1}{2}m_{n+1^+}^T Q_{n+1^+}m_{n+1^+} + \Phi(n+1, n+2)\bar{m}_{n+2}Q_{n+1^+}m_{n+2^+} \end{aligned} \tag{27}$$

Therefore, for Segment $n$ the value function at the terminal time $t_{n+1}$ is given by

$$\begin{aligned} V_{n+1} &= V_{n+1^+} + \frac{1}{2}(m_{n+1} - \bar{m}_{n+1})^T R_{n+1}(m_{n+1} - \bar{m}_{n+1}) \\ &= \frac{1}{2}m_{n+1}^\intercal(Q_{n+1^+} + R_{n+1})m_{n+1} + (\Phi(n+1, T)\bar{m}_T Q_{n+1^+} \\ &\quad - \bar{m}_{n+1}R_{n+1})m_{n+1} + const. \\ &= \frac{1}{2}m_{n+1}^T Q_{n+1}m_{n+1} + r_{n+1}Q_{n+1}m_{n+1} + const. \end{aligned} \tag{28}$$

This suggests that the terminal constraints for the Ricatti equation $Q_t$, and the reference dynamics vector $r_t$ are given by

$$Q_{n+1} := Q_{n+1^+} + R_{n+1} \tag{29}$$
$$r_{n+1} := (\Phi(t_{n+1}, T)\bar{m}_T Q_{n+1^+} - \bar{m}_{n+1}R_{n+1})Q_{n+1}^{-1} \tag{30}$$

Lastly, recall that the Lyapunov function is defined as $P_t = Q_t^{-1}$, thus the terminal constraints with respect to the Lyapunov function are given by

$$P_{n+1} = (P_{n+1^+}^{-1} + R_{n+1})^{-1}, \qquad r_{n+1} = P_{n+1}(P_{n+1^+}^{-1}r_{n+1^+} - R_{n+1}\bar{m}_{n+1}) \tag{31}$$

This implies that $r_t$—and hence the optimal control $u_t^\star$—would depend on all preceding pinned points after $t$ $\{\bar{m}_j : t_j \geq t\}$. Finally, notice that for the last segment $t \in [t_{N-1}, t_N]$, it holds that $P_{N^+} = 0$, since there is no effect from any subsequent segment, hence $P_{t_N} = R_{t_N}^{-1}$, and $r_{t_N}$ simplifies to $-m_{t_N}$. $\qquad\square$

## B.2 Proof of Proposition 3.2

**Proposition B.2.** *Let* $R = \begin{bmatrix} \frac{1}{c} & 0 \\ 0 & c \end{bmatrix}$. *At the limit when $c \to 0$, the conditional acceleration in Eq. 3 admits an analytic form:*

$$a_t^\star(m_t | \{\bar{x}_{n+1} : t_{n+1} \geq t\}) = C_1^n(t)(x_t - \bar{x}_{n+1}) + C_2^n(t)v_t + C_3^n(t) \sum_{j=n+1}^N \lambda_j \bar{x}_j, \ t \in [t_n, t_{n+1})$$

(32)

*where $\{\bar{x}_{n+1} : t_{n+1} \geq t\}$ signifies the bridge is conditioned on the set of the ensuing points, $\lambda_j$ are static coefficients and $C_1^n(t), C_2^n(t), C_3^n(t)$ are time-varying coefficients specific to each segment.*

*Proof.* We start our analysis with the last segment $N - 1$, and then move to derive the formulation for an arbitrary preceding segment $n$.

**Segment $N - 1$:** $t \in [t_{N-1}, T]$
For the last segment, the terminal constraint $t \in [t_{N-1}, T]$ is given by $P_N = R_N^{-1}$, and hence $r_N$ simplifies to $-m_N$. Solving the backward differential equation $P_t$, for $t \in [t_{N-1}, T]$, with $P_T = P_N = R_N^{-1}$, yields

$$P(t) = \begin{bmatrix} -\frac{\sigma^2}{2}(t-T)^3 + \frac{(t-T)^2}{c} + c & -\frac{\sigma^2}{2}(t-T)^2 + \frac{(t-T)}{c} \\ -\frac{\sigma^2}{2}(t-T)^2 + \frac{(t-T)}{c} & -\sigma^2(t-T) + \frac{1}{c} \end{bmatrix}$$

(33)

Additionally, solving for $r_t$ for $t \in [t_{N-1}, T]$, with $r_T = -\bar{m}_T$ yields:

$$r_t = \Phi(t, T)\bar{m}_T$$

(34)

where $\Phi(t, s)$ is the transition matrix of the following dynamics $dm_t = A(t)m_t dt$, and is defined as $\Phi(t, s) = \begin{bmatrix} 1 & t-s \\ 0 & 1 \end{bmatrix}$. Plugging Eq. 33, and 34 into Eq. 17 yields

$$u_t^\star = \begin{bmatrix} 0 \\ \frac{3}{(t-T)^2}(\bar{x}_T - x_t) - \frac{3}{T-t}v_t \end{bmatrix}, \forall t \in [t_{N-1}, T]$$

(35)

Therefore, regardless of the total number of marginals the $C$-functions for the last segments are always: $C_1^{(N-1)}(t) = -\frac{3}{(T-t)^2}, \quad C_2^{(N-1)}(t) = \frac{3}{T-t}, \quad C_3^{(N-1)}(t) = 0$.

**Segment $n$:** $t \in [t_n, t_{n+1}]$
Now, we move to derive the conditional acceleration, for an arbitrary segment $n$, with $n < N - 1$ i.e. $n$ is not the last segment. Let us recall the optimal control formulation from Eq. (17)

$$u_t^\star(m_t) = -\mathbf{g}\mathbf{g}^\mathsf{T} P_t^{-1}(m_t + \Phi(t, t_{n+1})r_{n+1})$$

(36)

For convenience, let us define the following functions corresponding to the $n^{\text{th}}$ segment: $t \in [t_n, t_{n+1})$

$$
\begin{aligned}
z_1^{(n)}(t) &= 3t - 3t_{n+1} - 3 & z_2^{(n)}(t) &= 3t - 4 - 3t_{n+1} \\
z_3^{(n)}(t) &= 6t - 6t_{n+1} - 3 & z_4^{(n)}(t) &= 6t - 6t_{n+1} - 4 \\
z_5^{(n)}(t) &= 4t - 4t_{n+1} - 3 & z_6^{(n)}(t) &= 4t - 4 - 4t_{n+1} \\
z_7^{(n)}(t) &= 6t - 6t_{n+1} + 3 & z_8^{(n)}(t) &= 6t - 6t_{n+1} + 4
\end{aligned}
$$

(37)

For this arbitrary segment $n$, we solve the backward ODE $P_t$, with the corresponding terminal, using an ODE solution software.

$$\dot{P}_t = AP_t + P_t A^\mathsf{T} - \sigma_t \sigma_t^\mathsf{T}, \qquad P_n = (P_{n^+}^{-1} + R)^{-1}, \text{ with } P_{N^+} = 0$$

(38)

Given the structure of $R = \begin{bmatrix} \frac{1}{c} & 0 \\ 0 & c \end{bmatrix}$, with $c \to 0$ this leads us to the following expression for $P_t$, for $t \in [t_n, t_{n+1})$

$$P_t = \begin{bmatrix} \sigma^2 \frac{(t-t_{n+1})^3(\alpha^{(n)}z_1^{(n)}(t)+\beta^{(n)}z_2^{(n)}(t))}{2(\alpha^{(n)}z_5^{(n)}(t)+\beta^{(n)}z_6^{(n)}(t)))} & \sigma^2 \frac{(t-t_{n+1})^2(\alpha^{(n)}z_1^{(n)}(t)+\beta^{(n)}z_2^{(n)}(t))}{2(\alpha^{(n)}z_3^{(n)}(t)+\beta^{(n)}z_4^{(n)}(t))} \\ \sigma^2 \frac{(t-t_{n+1})^2(\alpha^{(n)}z_1^{(n)}(t)+\beta^{(n)}z_2^{(n)}(t))}{2(\alpha^{(n)}z_3^{(n)}(t)+\beta^{(n)}z_4^{(n)}(t))} & 2\sigma^2 \frac{(t-t_{n+1})(\alpha^{(n)}z_1^{(n)}(t)+\beta^{(n)}z_2^{(n)}(t))}{(\alpha^{(n)}z_7^{(n)}(t)+\beta^{(n)}z_8^{(n)}(t))} \end{bmatrix}$$

(39)

where $\alpha^{(n)}$, $\beta^{(n)}$ are segment-specific coefficients that shape the conditional bridge for the given segment $n$. These are recursively computed using the coefficients of the subsequent segment $n + 1$, as follows:

$$\alpha^{(n)} = \alpha^{(n+1)} z_1^{(n)}(t_{n+1}) + \beta^{(n+1)} z_2^{(n)}(t_{n+1})$$
$$\beta^{(n)} = 4(\alpha^{(n+1)} + \beta^{(n+1)}) \tag{40}$$

From the expression of $P_t$, we can obtain its inverse

$$P_t^{-1} = \frac{1}{\sigma^2} \begin{bmatrix} \frac{6(\alpha^{(n)} z_7^{(n)}(t) + \beta^{(n)} z_8^{(n)}(t))}{(t-t_{n+1})^3 * (\alpha^{(n)} z_1^{(n)}(t) + \beta^{(n)} z_2^{(n)}(t))} & \frac{-3(\alpha^{(n)} z_3^{(n)}(t) + \beta^{(n)} z_4^{(n)}(t))}{(t-t_{n+1})^2 * (\alpha^{(n)} z_1^{(n)}(t) + \beta^{(n)} z_2^{(n)}(t))} \\ \frac{-3(\alpha^{(n)} z_3^{(n)}(t) + \beta^{(n)} z_4^{(n)}(t))}{(t-t_{n+1})^2 * (\alpha^{(n)} z_1^{(n)}(t) + \beta^{(n)} z_2^{(n)}(t))} & \frac{3(\alpha^{(n)} z_5^{(n)}(t) + \beta^{(n)} z_6^{(n)}(t))}{(t-t_{n+1}) * (\alpha^{(n)} z_1^{(n)}(t) + \beta^{(n)} z_2^{(n)}(t))} \end{bmatrix} \tag{41}$$

Hence, we can compute the terms: i) $\mathbf{gg}^{\mathsf{T}} P_t^{-1}$, ii) $\mathbf{gg}^{\mathsf{T}} P_t^{-1} \Phi(t, t_{n+1})$ in the optimal control formulation as follows:

$$\mathbf{gg}^{\mathsf{T}} P_t^{-1} = \begin{bmatrix} 0 & 0 \\ \frac{-3(\alpha^{(n)} z_3^{(n)}(t) + \beta^{(n)} z_4^{(n)}(t))}{(t-t_{n+1})^2 * (\alpha^{(n)} z_1^{(n)}(t) + \beta^{(n)} z_2^{(n)}(t))} & \frac{3(\alpha^{(n)} z_5^{(n)}(t) + \beta^{(n)} z_6^{(n)}(t))}{(t-t_{n+1}) * (\alpha^{(n)} z_1^{(n)}(t) + \beta^{(n)} z_2^{(n)}(t))} \end{bmatrix} \tag{42}$$

$$\mathbf{gg}^{\mathsf{T}} P_t^{-1} \Phi(t, t_{n+1}) = \begin{bmatrix} 0 & 0 \\ \frac{-3(\alpha^{(n)} z_3^{(n)}(t) + \beta^{(n)} z_4^{(n)}(t))}{(t-t_{n+1})^2 * (\alpha^{(n)} z_1^{(n)}(t) + \beta^{(n)} z_2^{(n)}(t))} & \frac{6(\alpha^{(n)} + \beta^{(n)})}{(\alpha^{(n)} z_1^{(n)}(t) + \beta^{(n)} z_2^{(n)}(t))} \end{bmatrix} \tag{43}$$

Therefore, we can rewrite the optimal control in Eq. (17) as

$$u_t^\star = \begin{bmatrix} 0 & 0 \\ C_1^{(n)}(t) & C_2^{(n)}(t) \end{bmatrix} \begin{bmatrix} \mathbf{x}_t \\ \mathbf{v}_t \end{bmatrix} + \begin{bmatrix} 0 & 0 \\ C_1^{(n)}(t) & C_3^{(n)}(t) \end{bmatrix} r_{n+1}, \forall t \in [t_n, t_{n+1}] \tag{44}$$

where we have that

$$C_1^{(n)}(t) = \frac{-3(\alpha^{(n)} z_3^{(n)}(t) + \beta^{(n)} z_4^{(n)}(t))}{(t - t_{n+1})^2 * (\alpha^{(n)} z_1^{(n)}(t) + \beta^{(n)} z_2^{(n)}(t))}$$
$$C_2^{(n)}(t) = \frac{3(\alpha^{(n)} z_5^{(n)}(t) + \beta^{(n)} z_6^{(n)}(t))}{(t - t_{n+1}) * (\alpha^{(n)} z_1^{(n)}(t) + \beta^{(n)} z_2^{(n)}(t))} \tag{45}$$
$$C_3^{(n)}(t) = \frac{6(\alpha^{(n)} + \beta^{(n)})}{(\alpha^{(n)} z_1^{(n)}(t) + \beta^{(n)} z_2^{(n)}(t))}$$

Now we proceed to the computation of the term $r_{n+1}$, for which it holds from Eq. (13), that

$$r_{n+1} = P_{n+1}(P_{n+1+}^{-1} r_{n+1+} - R \bar{m}_{n+1}) \tag{46}$$

where the term $r_{n+1+} = \Phi(t_{n+1}, t_{n+2}) r_{n+2}$ carries the impact from the future segments. Therefore, the first term recursively introduces the impact of further future pinned points through

$$P_{n+1} P_{n+1+}^{-1} r_{n+1+} = P_{n+1} P_{n+1+}^{-1} \Phi(t_{n+1}, t_{n+2}) r_{n+2} \tag{47}$$

Since $P_{n+1} P_{n+1+}^{-1} \Phi(t_{n+1}, t_{n+2})$ is also independent of time, we have that

$$P_{n+1} P_{n+1+}^{-1} \Phi(t_{n+1}, t_{n+2}) = \begin{bmatrix} 0 & 0 \\ \lambda^{(n)} & 0 \end{bmatrix} \tag{48}$$

where $\lambda$ is some static coefficient, specific for the $n^{\text{th}}$ segment, i.e., different segments are characterized by different $\lambda$ coefficients.

The structure of this matrix in Eq. (48) suggests that $r_t$ will be dependent only on the positional constraints, when multiplied with $\bar{m}_j$ for $j = \{n + 2, \ldots, N\}$. Finally, given the linearity of the dynamics of $r_t$, we can recursively add the impact of more pinned points

$$P_{n+1} P_{n+1+}^{-1} \Phi(t_{n+1}, t_{n+2}) r_{n+2} = P_{n+1} P_{n+1+}^{-1} \Phi(t_{n+1}, t_{n+2}) \big(P_{n+2}(P_{n+2+}^{-1} r_{n+2+} - R \bar{m}_{n+2})\big) \tag{49}$$

This recursion leads to $r_t$ being expressed as a linear combination of those future pinned points, through the following expression

$$P_{n+1}P_{n+1+}^{-1}r_{n+1+} = \begin{bmatrix} 0 \\ \sum_{j=n+2}^{N} \lambda_j^{(n)} \bar{x}_j \end{bmatrix} \tag{50}$$

Subsequently, computation of $P_{n+1}$ from $P_{n+1} = (P_{n+1+} + R)^{-1}$ along with the diagonal structure of $R$ also leads to

$$P_{n+1} R \bar{m}_{n+1} = \begin{bmatrix} -x_{n+1} \\ \kappa^{(n)} x_{n+1} \end{bmatrix} \tag{51}$$

where $\kappa^{(n)}$ is also some static coefficient, specific to the $n^{\text{th}}$ segment. Consequently, this leads to the following expression for $r_{n+1}$

$$r_{n+1} = \begin{bmatrix} -\bar{x}_{n+1} \\ \kappa^{(n)}\bar{x}_{n+1} + \sum_{j=n+2}^{N} \lambda_j^{(n)} \bar{x}_j \end{bmatrix} \tag{52}$$

or equivalently

$$r_{n+1} = \begin{bmatrix} -\bar{x}_{n+1} \\ \sum_{j=n+1}^{N} \lambda_j^{(n)} \bar{x}_j \end{bmatrix} \tag{53}$$

where we defined $\kappa^{(n)} = \lambda_{n+1}^{(n)}$. This implies that $r_t$—and hence the optimal control $u_t^\star$— depends on all preceding pinned points after $t$ $\{\bar{m}_{n+1} : t_{n+1} \geq t\}$, as a linear combination of these points. It is found that the elements of the vector $\lambda^{(n)} = [\lambda_{n+1}, \lambda_{n+2}, \ldots, \lambda_N]$ depend only on the number of accounted pinned points $\bar{x}_j$, and decay exponentially as this number increases, as illustrated in Figure 10. In other words, the values of $\lambda_j^{(n)}$ decrease the further the corresponding $\bar{x}_j$ is located from the segment whose bridge we compute.

*Remark* B.3. It is highlighted that the sole dependency of the coefficients $\alpha^{(n)}, \beta^{(n)}, \lambda^{(n)}$ is the number of future marginals.

$\square$

### B.2.1  Examples of Multi-Marginal Bridges

At this point, we provide examples of multi-marginal conditional bridges, elucidating that the structure of each segment is governed by the number of future marginals. For simplicity, let us denote with $\alpha^{(n)}, \beta^{(n)}, \lambda^{(n)}$ the segment-specific coefficients corresponding to the $n^{\text{th}}$ segment: $t \in [t_n, t_{n+1})$.

**2-marginal Bridge**  The formulation for a two-marginal bridge coincides with the segment $N-1$ for a multi-marginal bridge, when $N = 2$, and $T = 1$. More specifically, we have:

$$a^\star(m_t|, \bar{x}_1) = \frac{3}{(t-1)^2}(\bar{x}_T - x_t) - \frac{3}{1-t}v_t, \forall t \in [0,1) \tag{54}$$

**3-marginal Bridge**  The formulation of the 3-marginal bridge is given by

$$\begin{cases} a^\star(m_t|, \bar{x}_2) = \dfrac{3}{(T-t)^2}(\bar{x}_2 - x_t) - \dfrac{3}{T-t}v_t, & t \in [t_1, T) \\[3mm] a^\star(m_t|\bar{x}_1, \bar{x}_2) = \dfrac{-18t - 18t_1 - 12}{(t-1)^2(3t - 3t_1 - 4)}(x_t - \bar{x}_1) + \dfrac{12t - 12t_1 - 12}{(t-1)(3t - 3t_1 - 4)}v_t \\[3mm] \qquad + \dfrac{6}{(3t - 3t_1 - 4)}(\bar{x}_2 - \bar{x}_1), & t \in [t_0, t_1) \end{cases} \tag{55}$$

Notice that for $t_1 = 1$, and $t_2 = 2$, we derive the same expression as in the Example in Section 3. Additionally, we see that the segment $t \in [t_0, t_1)$ is obtained by our generalized formula for $\alpha = 0$, $\beta = 1$, and coincides with the expression of the $N-2$ segment. Finally, it is verified that the last segment shares the same formulation with the same coefficients as the bridge of the 2-marginal case.

**4-marginal Bridge**  The 4-marginal bridge further illustrates how the structure of each segment depends on the number of future marginals. In particular, following Remark B.3, the last two segments $t \in [t_2, T)$ and $t \in [t_1, t_2)$ share the same formulation as in the 3-marginal bridge, since they are conditioned on 1 and 2-future marginals, respectively. To compute the first segment, $t \in [t_0, t_1)$, we find that $\alpha^{(0)} = 4$, $\beta^{(0)} = 4$, and the $\lambda^{(0)}$ vector to be: $\lambda^{(0)} = [-1.25, 1.5, -0.25]^{\mathsf{T}}$. This results in the following bridge formulation:

$$
\begin{cases}
a^\star(m_t|, \bar{x}_3) = \dfrac{3}{(T-2)^2}(\bar{x}_3 - x_t) - \dfrac{3}{T-t}v_t, & t \in [t_2, T) \\[4mm]
a^\star(m_t|\bar{x}_2, \bar{x}_3) = \dfrac{-18t - 18t_2 - 12}{(t-t_2)^2(3t - 3t_2 - 4)}(x_t - \bar{x}_2) & \\[2mm]
\qquad + \dfrac{12t - 12t_2 - 12}{(t-t_2)(3t - 3t_2 - 4)}v_t + \dfrac{6}{(3t - 3t_2 - 4)}(\bar{x}_3 - \bar{x}_2), & t \in [t_1, t_2) \\[4mm]
a^\star(m_t|\bar{x}_1, \bar{x}_2, \bar{x}_3) = \dfrac{-36t + 36t_1 - 21}{(t-t_1)^2(6t - 6t_1 - 7)}(x_t - \bar{x}_1) + \dfrac{24t - 24t_1 - 21}{(t-t_1)(6t - 6t_1 - 7)}v_t & \\[2mm]
\qquad + \dfrac{12}{(6t - 6t_1 - 7)}(-0.25\bar{x}_3 + 1.5\bar{x}_2 - 1.25\bar{x}_1) & t \in [t_0, t_1)
\end{cases}
$$

$$(56)$$

**5-marginal Bridge**  It is easy to see that the last three segments of the 5-marginal bridge follow the same structure as in the 4-marginal case. For example, based on Remark B.3, the coefficients for the third-to-last segment, $t \in [t_1, t_2)$, are $\alpha_1 = \beta_1 = 4$, and the vectors $\lambda^{(2)}$ and $\lambda^{(1)}$, corresponding to the segments $t \in [t_2, t_3)$ and $t \in [t_1, t_2)$, respectively, match those of the third-to-last and second-to-last segments in the 4-marginal bridge. Finally, for the first segment, $t \in [0, t_1)$, we compute that: $\alpha^{(0)} = 28$, $\beta^{(0)} = 32$, and $\lambda^{(0)} = [-1.267, 1.6, -0.4, 0.067]^{\mathsf{T}}$. Substituting these coefficients into Eq. (45) yields the corresponding bridge formulation.

$$
\begin{cases}
a^\star(m_t \mid \bar{x}_4) = \dfrac{3}{(T-t)^2}(\bar{x}_4 - x_t) - \dfrac{3}{T-t}v_t, & t \in [t_3, T) \\[4mm]
a^\star(m_t \mid \bar{x}_3, \bar{x}_4) = \dfrac{-18t - 18t_3 - 12}{(t-t_3)^2(3t - 3t_3 - 4)}(x_t - \bar{x}_3) & \\[2mm]
\qquad + \dfrac{12t - 12t_3 - 12}{(t-t_3)(3t - 3t_3 - 4)}v_t + \dfrac{6}{3t - 3t_3 - 4}(\bar{x}_4 - \bar{x}_3), & t \in [t_2, t_3) \\[4mm]
a^\star(m_t \mid \bar{x}_2, \bar{x}_3, \bar{x}_4) = \dfrac{-36t + 36t_2 - 21}{(t-t_2)^2(6t - 6t_2 - 7)}(x_t - \bar{x}_2) + \dfrac{24t - 24t_2 - 21}{(t-t_2)(6t - 6t_2 - 7)}v_t & \\[2mm]
\qquad + \dfrac{12}{(6t - 6t_1 - 7)}(-0.25\bar{x}_4 + 1.5\bar{x}_3 - 1.25\bar{x}_2), & t \in [t_1, t_2) \\[4mm]
a^\star(m_t \mid \bar{x}_1, \bar{x}_2, \bar{x}_3, \bar{x}_4) = -\dfrac{6\,(45t - 45t_1 - 26)}{(t-t_1)^2\,(45t - 45t_1 - 52)}(x_t - \bar{x}_1) + \dfrac{12\,(15t - 15t_1 - 13)}{(t-t_1)\,(45t - 45t_1 - 52)}v_t & \\[2mm]
\qquad + \dfrac{90}{45t - 45t_1 - 52}(-1.267\bar{x}_1 + 1.6\bar{x}_2 - 0.4\bar{x}_3 + 0.067\bar{x}_4), & t \in [t_0, t_1)
\end{cases}
$$

$$(57)$$

### B.3  Proof of Proposition 3.4

**Proposition B.4.** *Let us define the marginal path $p_t$ as a mixture of bridges $p_t(m_t) = \int p_{t|\{\bar{x}_n\}}(m_t|\{\bar{x}_n : t_n \geq t\})dq(\{x_n\})$, where $p_{t|\{\bar{x}_n\}}(m_t|\{\bar{x}_n : t_n \geq t\})$ is the conditional probability path associated with the solution of the 3MBB path in Eq 6. The parameterized acceleration*

*that satisfies the FPE prescribed by the $p_t$ is given by*

$$a_t(t, m_t) = \frac{\int a_{t|\{\bar{x}_n\}} p_{t|\{\bar{x}_n\}}(m_t|\{\bar{x}_n : t_n \geq t\}) dq(\{x_n\})}{p_t} \tag{58}$$

*This suggests that the minimization of the variational gap to match $a_t^\theta$ given $p_t$ is given by*

$$\min_\theta \mathbb{E}_{q(\{x_n\})} \mathbb{E}_{p_{t|\{\bar{x}_n\}}} \big[ \int_0^1 \|a_{t|\{\bar{x}_n\}} - a_t^\theta\|^2 dt \big] \tag{59}$$

*Proof.* We want to show that the acceleration from Eq. 9 preserves the prescribed path $p_t$. The momentum Fokker Plank Equation (FPE) is given by

$$\partial_t p_t(m_t) = -v_t \nabla_x p_t(m_t) - \nabla_v(a_t(m_t) p_t(m_t)) + \frac{1}{2}\sigma^2 \Delta_v p_t(m_t) \tag{60}$$

We let the marginal be constructed as a mixture of conditional probability paths conditioned on a collection of pinned points $\{\bar{x}_n\}_{n\in[0,N]}$, $p_t = \int p_t(m_t|\{\bar{x}_n\}) q(\{\bar{x}_n\}) dx_0 dx_1 \dots dx_N$. Using this definition for the marginal path, one obtains that

$$\partial_t p_t(m_t) = \partial_t \int q(\{\bar{x}_n\}) p_t(m_t|\{\bar{x}_n\}) d\{\bar{x}_n\} = \int q(\{\bar{x}_n\}) \partial_t p_t(m_t|\{\bar{x}_n\}) d\{\bar{x}_n\}$$

$$v_t \nabla_x p_t(m_t) = v_t \nabla_x \int q(\{\bar{x}_n\}) p_t(m_t|\{\bar{x}_n\}) d\{\bar{x}_n\} = \int q(\{\bar{x}_n\}) \big[ v_t \nabla_x p_t(m_t|\{\bar{x}_n\}) \big] d\{\bar{x}_n\}$$

$$\Delta_v p_t(m_t) = \nabla_v \cdot (\nabla_v p_t(m_t)) = \nabla_v \cdot (\nabla_v \int q(\{\bar{x}_n\}) p_t(m_t|\{\bar{x}_n\}) d\{\bar{x}_n\})$$

$$= \int q(\{\bar{x}_n\}) \Big[ \nabla_v \cdot (\nabla_v \int p_t(m_t|\{\bar{x}_n\})) \Big] d\{\bar{x}_n\})$$

$$= \int q(\{\bar{x}_n\}) \Delta_v p_t(m_t|\{\bar{x}_n\}) d\{\bar{x}_n\}$$

$$\tag{61}$$

Hence it remains to be checked whether the following equality holds

$$a_t(m_t) p_t(m_t) = \int q(\{\bar{x}_n\}) \big[ a_{t|1}(m_t|\{\bar{x}_n\}) p_t(m_t|\{\bar{x}_n\}) \big] d\{\bar{x}_n\} \tag{62}$$

which suggests that the parameterized drift that minimizes the following minimization problem

$$a_t^{\theta^\star}(m_t) = \arg\min_\theta \mathbb{E}_{q(\{\bar{x}_n\}) p_t(m_t|\{\bar{x}_n\})} \big[ \|a_t^\theta(m_t) - a_{t|\bar{x}_n}(m_t|\{\bar{x}_n\})\|^2 \big] \tag{63}$$

preserves the prescribed $p_t$. $\qquad\square$

## B.4 Proof of Theorem 3.5

**Definition B.5** (Markovian Projection of path measure)**.** The Markovian Projection of $\mathbb{P}$ is defined as $\mathbb{P}^{\mathcal{M}} = \arg\min_{\mathbb{V}\in\mathcal{M}} KL(\mathbb{P}|\mathbb{V})$.

Intuitively, the Markovian Projection seeks the path measure that minimizes the variational distance to $\mathbb{P}$. In other words, it seeks the closest Markovian path measure in the KL sense.

**Definition B.6** (Reciprocal Class and Projection)**.** For multi-marginal path measures, we say that $\mathbb{P}$ is in the reciprocal class $\mathcal{R}(\mathbb{Q})$ of $\mathbb{Q} \in \mathcal{M}$ if

$$\mathbb{P} = \int \mathbb{Q}_{|\{x_n\}} dq(\{x_n\})$$

namely, it shares the same bridges with $\mathbb{Q}$. We define the *reciprocal projection* of $\mathbb{P}$ as

$$\mathbb{P}^\star = \text{proj}_{\mathcal{R}(\mathbb{Q})}(\mathbb{P}) := \arg\min_{\mathbb{T}\in\mathcal{R}(\mathbb{Q})} KL(\mathbb{P} \| \mathbb{T}).$$

Similarly, the Reciprocal Projection yields the closest reciprocal path measure in the KL sense.

**Lemma B.7.** *Let $\mathbb{P}$ be a Markov measure in the reciprocal class of $\mathbb{Q} \in \mathcal{M}$ such that $\int \mathbb{P}_n dv_n = q_n(x_n)$, for $n \in \{0, \ldots, N\}$. Then, under some mild regularity assumptions on $\mathbb{Q}$, $q_n$, it is found that $\mathbb{P}$ is equal to the unique multi-marginal the Schrödinger Bridge $\mathbb{P}^{mmSB}$.*

*Proof.* First let us assume that $\mathrm{KL}(\mathbb{P}|\mathbb{Q}) < \infty$, and that $\mathrm{KL}(q_n | \int \mathbb{Q}_n dv_n) < \infty$ for $n \in \{0, \ldots, N\}$. Assume $\mathbb{Q} \in \mathcal{M}$, then by (Theorem 2.10, Theorem 2.12 Léonard [2013]), it follows that the solution of the dynamic SB $\mathbb{P}$ must also be a Markov measure. Finally, from the factorization of the KL, it holds that

$$\mathrm{KL}(\mathbb{P}|\mathbb{Q}) = \mathrm{KL}(\mathbb{P}_{\{x_n\}}|\mathbb{Q}_{\{x_n\}}) + \int \mathrm{KL}(\mathbb{P}_{|\{x_n\}}|\mathbb{Q}_{\{x_n\}})d\mathbb{P}_{\{x_n\}} \tag{64}$$

which implies that $\mathrm{KL}(\mathbb{P}_{\{x_n\}} \mid \mathbb{Q}_{\{x_n\}}) \leq \mathrm{KL}(\mathbb{P} \mid \mathbb{Q})$ with equality (when $\mathrm{KL}(\mathbb{P} \mid \mathbb{Q}) < \infty$) if and only if $\mathbb{P}_{|\{x_n\}} = \mathbb{Q}_{|\{x_n\}}$. Therefore, $\mathbb{P}^\star$ is the (unique) solution mmSB if and only if it disintegrates as above (Proposition 2.3 Léonard [2013]). $\qquad\square$

**Lemma B.8.** *[Proposition 6 in [Shi et al., 2023]] Let $\mathbb{V} \in \mathcal{M}$ and $\mathbb{T} \in \mathcal{R}(\mathbb{Q})$ and . If $\mathrm{KL}(\mathbb{P}|\mathbb{V}) < \infty$, and if $\mathrm{KL}(\mathrm{proj}_{\mathcal{M}}(\mathbb{P})|\mathbb{V}) < \infty$ we have*

$$\mathrm{KL}(\mathbb{P}|\mathbb{V}) = \mathrm{KL}(\mathbb{P}| \mathrm{proj}_{\mathcal{M}}(\mathbb{P})) + \mathrm{KL}(\mathrm{proj}_{\mathcal{M}}(\mathbb{P})|\mathbb{V}). \tag{65}$$

*and if $\mathrm{KL}(\mathbb{P}|\mathbb{T}) < \infty$, then*

$$\mathrm{KL}(\mathbb{P}|\mathbb{T}) = \mathrm{KL}(\mathbb{P}| \mathrm{proj}_{\mathcal{R}(\mathbb{Q})}(\mathbb{P})) + \mathrm{KL}(\mathrm{proj}_{\mathcal{R}(\mathbb{Q})}(\mathbb{P})|\mathbb{T}). \tag{66}$$

**Theorem B.9.** *Assume that the conditions of Lemma B.7, and B.8 hold. Then, our iterative scheme admits a fixed point solution $\mathbb{P}^\star$, i.e., $\mathrm{KL}(\mathbb{P}^i|\mathbb{P}^\star) \to 0$, and in particular, this fixed point coincides with the unique $\mathbb{P}^{mmSB}$.*

*Proof.* We define the following path measures $\mathbb{V} = \mathrm{proj}_{\mathcal{M}}(\mathbb{P})$, and $\mathbb{T} = \mathrm{proj}_{\mathcal{R}(\mathbb{Q})}(\mathbb{V})$. Assume the conditions for Lemma B.8 hold for $\mathbb{P}$, $\mathbb{V}$ and $\mathbb{T}$. Then for any arbitrary fixed point $\mathbb{V}'$, we can write

$$\mathrm{KL}(\mathbb{V}^0|\mathbb{V}') = \mathrm{KL}(\mathbb{V}^0|\mathbb{V}^1) + \mathrm{KL}(\mathbb{V}^1|\mathbb{V}') = \sum_{i=0}^{N} \mathrm{KL}(\mathbb{V}^i|\mathbb{V}') + \mathrm{KL}(\mathbb{V}^i|\mathbb{V}') \tag{67}$$

Thus, it holds that $\mathrm{KL}(\mathbb{V}^i|\mathbb{V}') < \infty$ for every iteration $i \in \mathbb{N}$. Similarly, for $\mathbb{P}$, and $\mathbb{T}$, we obtain $\mathrm{KL}(\mathbb{P}^i|\mathbb{P}') < \infty$ and $\mathrm{KL}(\mathbb{T}^i|\mathbb{T}') < \infty$ for each $i \in \mathbb{N}$ for any arbitrary fixed $\mathbb{P}'$ and $\mathbb{T}'$.

Consequently, we define the following function

$$\Psi^i := \mathrm{KL}(\mathbb{V}^i|\mathbb{V}') + \mathrm{KL}(\mathbb{T}^i|\mathbb{T}') + \mathrm{KL}(\mathbb{P}^i|\mathbb{P}'). \tag{68}$$

For two consecutive iterates $i$ and $i + 1$, we have

- $\Psi^i := \mathrm{KL}(\mathbb{V}^i|\mathbb{V}') + \mathrm{KL}(\mathbb{T}^i|\mathbb{T}') + \mathrm{KL}(\mathbb{P}^i|\mathbb{P}')$

- $\Psi^{i+1} := \mathrm{KL}(\mathbb{V}^{i+1}|\mathbb{V}') + \mathrm{KL}(\mathbb{T}^{i+1}|\mathbb{T}') + \mathrm{KL}(\mathbb{P}^i|\mathbb{P}')$

Using Lemma B.8, we can rewrite $\Psi^i$ as

$$\Psi^i := \mathrm{KL}(\mathbb{V}^i|\mathbb{V}') + \mathrm{KL}(\mathbb{T}^i|\mathbb{T}') + \mathrm{KL}(\mathbb{P}^i|\mathbb{P}')$$
$$= \mathrm{KL}(\mathbb{V}^i|\mathbb{V}^{i+1}) + \mathrm{KL}(\mathbb{T}^i|\mathbb{T}^{i+1}) + \mathrm{KL}(\mathbb{P}^i|\mathbb{P}^{i+1}) + \mathrm{KL}(\mathbb{V}^{i+1}|\mathbb{V}') + \mathrm{KL}(\mathbb{T}^{i+1}|\mathbb{T}') + \mathrm{KL}(\mathbb{P}^{i+1}|\mathbb{P}')$$
$$= \mathrm{KL}(\mathbb{V}^i|\mathbb{V}^{i+1}) + \mathrm{KL}(\mathbb{T}^i|\mathbb{T}^{i+1}) + \mathrm{KL}(\mathbb{P}^i|\mathbb{P}^{i+1}) + \Psi^{i+1}$$

Now, we take the sum of this telescoping series and obtain

$$\Psi^0 - \Psi^\infty \geq \sum_{i=0}^{\infty} \mathrm{KL}(\mathbb{V}^i|\mathbb{V}^{i+1}) + \mathrm{KL}(\mathbb{T}^i|\mathbb{T}^{i+1}) + \mathrm{KL}(\mathbb{P}^i|\mathbb{P}^{i+1}) \tag{69}$$

Note that $\Psi^0$ and $\Psi^\infty$ are finite (with $\Psi^0 \geq \Psi^\infty$), since $\mathrm{KL}(\mathbb{P}^i|\mathbb{P}') < \infty$, $\mathrm{KL}(\mathbb{V}^i|\mathbb{V}') < \infty$ and $\mathrm{KL}(\mathbb{T}^i|\mathbb{T}^\star) < \infty$ for every iteration $i \in \mathbb{N}$. Therefore, since we also have $\mathrm{KL}(\mathbb{P}^i|\mathbb{P}^{i+1}) \geq 0$, $\mathrm{KL}(\mathbb{V}^i|\mathbb{V}^{i+1}) \geq 0$ and $\mathrm{KL}(\mathbb{T}^i|\mathbb{T}^{i+1}) \geq 0$, we get that

- $\lim_{i \to \infty} \text{KL}(\mathbb{P}^i | \mathbb{P}^{i+1}) \to 0,$

- $\lim_{i \to \infty} \text{KL}(\mathbb{V}^i | \mathbb{V}^{i+1}) \to 0,$

- $\lim_{i \to \infty} \text{KL}(\mathbb{T}^i | \mathbb{T}^{i+1}) \to 0.$

Hence the iterates $\mathbb{P}^i$, $\mathbb{V}^i$, and $\mathbb{T}^i$ converge to some fixed points $\mathbb{P}^i \to \mathbb{P}^\star$, $\mathbb{V}^i \to \mathbb{V}^\star$, and $\mathbb{T}^i \to \mathbb{T}^\star$, as $i \to \infty$.

From the factorization of the KL divergence, we have for consecutive projections of our algorithm:

$$
\begin{aligned}
\text{KL}(\mathbb{T}^{(i)} | \mathbb{P}^\star) &= \text{KL}(\mathbb{T}^{(i)}_{\{x_n\}} | \mathbb{P}^\star_{\{x_n\}}) + \mathbb{E}_{\mathbb{T}^{(i)}_{\{x_n\}}} \left[ \text{KL}(\mathbb{T}^{(i)}_{\{x_n\}} | \mathbb{P}^\star_{\{x_n\}}) \right] \\
&= \text{KL}(\mathbb{T}^{(i)}_{\{x_n\}} | \mathbb{P}^\star_{\{x_n\}})
\end{aligned}
\tag{70}
$$

since the bridges of $\mathbb{T}$ are the same with $\mathbb{P}$, and $\mathbb{Q}$, i.e., $\mathbb{T}^{(i)}_{|\{x_n\}} = \mathbb{P}^\star_{|\{x_n\}} = \mathbb{Q}_{|\{x_n\}}$. Then,

$$
\begin{aligned}
\text{KL}(\mathbb{V}^i | \mathbb{P}^\star) &= \text{KL}(\mathbb{V}^i_{\{x_n\}} | \mathbb{P}^\star_{\{x_n\}}) + \mathbb{E}_{\mathbb{V}^i_{\{x_n\}}} \left[ \text{KL}(\mathbb{V}^{(i)}_{\{x_n\}} | \mathbb{P}^\star_{\{x_n\}}) \right] \\
&\geq \text{KL}(\mathbb{T}^{(i+1)}_{\{x_n\}} | \mathbb{P}^\star_{\{x_n\}})
\end{aligned}
\tag{71}
$$

since the coupling after the Markovian projection at iteration $i$ remains the same for the reciprocal path measure at iteration $i + 1$, namely $\mathbb{V}^{(i)}_{\{x_n\}} = \mathbb{T}^{(i+1)}_{\{x_n\}}$. Therefore, we can deduce

$$
\text{KL}(\mathbb{V}^i | \mathbb{P}^\star) \geq \text{KL}(\mathbb{T}^{i+1} | \mathbb{P}^\star).
\tag{72}
$$

We further assume that $\text{KL}(\mathbb{T}^0 | \mathbb{P}^\star) < \infty$, $\text{KL}(\mathbb{V}^0 | \mathbb{P}^\star) < \infty$. Therefore, the iterations of Eq. 70 and 71, yield $\text{KL}(\mathbb{T}^{(i)} | \mathbb{P}^\star) \geq \text{KL}(\mathbb{V}^i | \mathbb{P}^\star) \geq \text{KL}(\mathbb{T}^{i+1} | \mathbb{P}^\star)$ for $i \geq 0$, implying convergence, since it is non-increasing and bounded below. Applying Lemma B.8, we obtain $\lim_{i \to \infty} \left( \text{KL}(\mathbb{T}^i | \mathbb{P}^\star) - \text{KL}(\mathbb{V}^i | \mathbb{P}^\star) \right) = \lim_{i \to \infty} \text{KL}(\mathbb{T}^i | \mathbb{V}^i) = 0$. By definition of the lower semi-continuity of the KL divergence, we have $\text{KL}(\mathbb{V}^* | \mathbb{T}^*) \leq \liminf_{k \to \infty} \text{KL}(\mathbb{V}^{i_{j^k}} | \mathbb{T}^{i_{j^k}})$. Additionally, by the definition of the KL divergence, we have $0 \leq \text{KL}(\mathbb{V}^* | \mathbb{T}^*)$. Finally, it also holds that $\liminf_{k \to \infty} \text{KL}(\mathbb{V}^{i_{j^k}} | \mathbb{T}^{i_{j^k}}) = 0$. Combining all three claims we have

$$
0 \leq \text{KL}(\mathbb{V}^* | \mathbb{T}^*) \leq \liminf_{k \to \infty} \text{KL}(\mathbb{V}^{i_{j^k}} | \mathbb{T}^{i_{j^k}}) = 0.
$$

Therefore, $\mathbb{V}^\star = \mathbb{T}^\star$, which also means that $\mathbb{P}^\star \in \mathcal{M} \cap \mathcal{R}(\mathbb{Q})$ [Shi et al., 2023]. Also, by construction, all the iterates of $\mathbb{P}^i$ satisfy the positional marginal constraints $\int \mathbb{P}^i_n dv_n = q_n$, hence also $\int \mathbb{P}^\star_n dv_n = q_n$. Therefore, by Lemma B.7, $\mathbb{P}^*$ is the unique multi-marginal Schrödinger bridge $\mathbb{P}^{\text{mmSB}}$.

$\square$

## C  Gaussian Path

The scalability of our framework is based on the capacity to efficiently sample from the conditional gaussian path induced by the solution of of the 3MBB path in Eq 6. Let us define the marginal path $p_t$ as a mixture of bridges $p_t(m_t) = \int p_{t|\{\bar{x}_n\}}(m_t | \{\bar{x}_n : t_n \geq t\}) dq(\{x_n\})$, where $p_{t|\{\bar{x}_n\}}(m_t | \{\bar{x}_n : t_n \geq t\})$ is the conditional probability path associated with the solution of the 3MBB path in Eq 6. The linearity of the system implies that we can efficiently sample $m_t = [x_t, v_t], \forall t \in [0, T]$, from the conditional probability path $p_{t|\{\bar{x}_n\}} = \mathcal{N}(\mu_t, \Sigma_t)$, as the mean vector $\mu_t$ and the covariance matrix $\Sigma_t$ have analytic solutions [Särkkä and Solin, 2019].

Let us recall the optimal control formulation for the $n^{\text{th}}$ segment.

$$
u_t^\star = \begin{bmatrix} 0 & 0 \\ C_1^{(n)}(t) & C_2^{(n)}(t) \end{bmatrix} \begin{bmatrix} \mathbf{x}_t \\ \mathbf{v}_t \end{bmatrix} + \begin{bmatrix} 0 & 0 \\ C_1^{(n)}(t) & C_3^{(n)}(t) \end{bmatrix} \begin{bmatrix} -\bar{x}_{n+1} \\ \sum_{j=n+1}^N \lambda_j \bar{x}_j \end{bmatrix}, \forall t \in [t_n, t_{n+1}]
\tag{73}
$$

Thus, the augmented SDE for the corresponding is written as

$$dm_t = \left( \begin{bmatrix} 0 & 1 \\ C_1^{(n)}(t) & C_2^{(n)}(t) \end{bmatrix} \begin{bmatrix} \mathbf{x}_t \\ \mathbf{v}_t \end{bmatrix} + \begin{bmatrix} 0 \\ -C_1^{(n)}(t)\bar{x}_{n+1} + (C_3^{(n)}(t)) \sum_{j=n+1}^{N} \lambda_j^{(n)} \bar{x}_j \end{bmatrix} \right) dt + \mathbf{g} dW_t \tag{74}$$

where $C_1^{(n)}(t), C_2^{(n)}(t), C_3^{(n)}(t)$ are the segment specific functions, and $\lambda_j^{(n)}$ the coefficients for the future pinned points. To find the expressions for the mean and the covariance, we follow [Särkkä and Solin, 2019], and consider the following ODEs for $\mu_t$, and $\Sigma_t$ respectively:

$$\dot{\mu}_t = \begin{bmatrix} 0 & 1 \\ C_1^{(n)}(t) & C_2^{(n)}(t) \end{bmatrix} \mu_t + \begin{bmatrix} 0 \\ -C_1^{(n)}(t)\bar{x}_{n+1} + C_3^{(n)}(t) \sum_{j=n+1}^{N} \lambda_j^{(n)} \bar{x}_j \end{bmatrix}$$

$$\dot{\Sigma}_t = \begin{bmatrix} 0 & 1 \\ C_1^{(n)}(t) & C_2^{(n)}(t) \end{bmatrix} \Sigma_t + [\begin{bmatrix} 0 & 1 \\ C_1^{(n)}(t) & C_2^{(n)}(t) \end{bmatrix} \Sigma_t]^\mathsf{T} + \mathbf{g}\mathbf{g}^\mathsf{T} \tag{75}$$

**Mean ODEs**   If we explicitly write the mean ODE system, we obtain the following two ODEs

$$\dot{\mu}_x = \mu_v$$

$$\dot{\mu}_v = C_1^{(n)}(t)\mu_x + C_2^{(n)}(t)\mu_v - C_1^{(n)}(t)\bar{x}_{n+1} + C_3^{(n)}(t) \sum_{j=n+1}^{N} \lambda_j^{(n)} \bar{x}_j \tag{76}$$

which corresponds to the following second-order ODE

$$\ddot{\mu}_x - C_2^{(n)}(t)\dot{\mu}_x + C_1^{(n)}(t)\mu_x = C_1^{(n)}(t)\bar{x}_{n+1} + C_3^{(n)}(t) \sum_{j=n+1}^{N} \lambda_j^{(n)} \bar{x}_j \tag{77}$$

This ODE is then solved using an ODE solver software for the corresponding functions $C^{(n)}$ of the respective segment $n$: $t \in [t_n, t_{n+1})$.

**Covariance ODEs**   If we explicitly write the mean ODE system, we obtain the following two ODEs

$$\dot{\Sigma}_{xx} = 2\Sigma_{xv}$$

$$\dot{\Sigma}_{xv} = C_1^{(n)}(t)\Sigma_{xx} + C_2^{(n)}(t)\Sigma_{xv} + \Sigma vv \tag{78}$$

$$\dot{\Sigma}_{vv} = 2C_1^{(n)}(t)\Sigma_{xv} + 2C_2^{(n)}(t)\Sigma_{vv} + \sigma^2$$

which corresponds to the following third-order ODE

$$\frac{1}{2}\dddot{\Sigma}_{xx} - \frac{3}{2}C_2^{(n)}(t)\ddot{\Sigma}_{xx} + (C_2^{(n)}(t)^2 - 2C_1^{(n)}(t) - \frac{1}{2}\dot{C}_2^{(n)}(t))\dot{\Sigma}_{xx} + (2C_2^{(n)}(t)C_1^{(n)}(t) - \dot{C}_1^{(n)}(t))\Sigma_{xx} = 0 \tag{79}$$

This equation, however, is hard to solve even using software packages. For this reason, we integrate the covariance ODEs using Euler integration, once at the beginning of our training. This procedure can be solved once and can be applied for any fixed coupling, during the matching, since the system of ODEs in Eq. (78) does not depend on any points $\bar{m}_n$, but its sole dependence is on time. This suggests that the computational overhead from simulating the covariance ODEs is negligible.

Given the expressions of $\mu_t = \begin{bmatrix} \mu_t^x \\ \mu_t^v \end{bmatrix}$, and $\Sigma_t = \begin{bmatrix} \Sigma_t^{xx} & \Sigma_t^{xv} \\ \Sigma_t^{xv} & \Sigma_t^{vv} \end{bmatrix}$, one can obtain $m_t$ through

$$m_t = \begin{bmatrix} X_t \\ V_t \end{bmatrix} = \begin{bmatrix} \mu_t^x \\ \mu_t^v \end{bmatrix} + \begin{bmatrix} L_t^{xx}\epsilon_0 \\ L_t^{xv}\epsilon_0 + L_t^{vv}\epsilon_1 \end{bmatrix} \tag{80}$$

where the matrix $L_t = \begin{bmatrix} L_t^{xx} & L_t^{xv} \\ L_t^{xv} & L_t^{vv} \end{bmatrix}$ is computed following the Cholesky decomposition to the covariance matrix, and $\epsilon = \begin{bmatrix} \epsilon_0 \\ \epsilon_1 \end{bmatrix} \sim \mathcal{N}(0, \mathbf{I}_{2d})$.

# D   Extended Related Works

**Schrödinger Bridge**   Recently, in generative modeling there has been a surge of principled approaches that stem from Optimal Transport [Villani et al., 2009]. The most prominent problem formulation has been the Schrödinger Bridge (SB; Schrödinger [1931]). In particular, SB gained significant popularity in the realm of generative modeling following advancements proposing a training scheme based on the Iterative Proportional Fitting (IPF), a continuous state space extension of the Sinkhorn algorithm to solve the dynamic SB problem [De Bortoli et al., 2021, Vargas et al., 2021]. Notably, SB generalizes standard diffusion models transporting data between arbitrary distributions $\pi_0, \pi_1$ with fully nonlinear stochastic processes, seeking the unique path measure that minimizes the kinetic energy. The Schrödinger Bridge [Schrödinger, 1931] in the path measure sense is concerned with finding the optimal measure $\mathbb{P}^\star$ that minimizes the following optimization problem

$$\min_{\mathbb{P}} \mathrm{KL}(\mathbb{P}|\mathbb{Q}), \quad \mathbb{P}_0 = \pi_0, \ \mathbb{P}_1 = \pi_1 \tag{81}$$

where $\mathbb{Q}$ is a Markovian reference measure. Hence the solution of the dynamic SB $\mathbb{P}^\star$ is considered to be the closest path measure to $\mathbb{Q}$. Another formulation of the dynamic SB crucially emerges by applying the Girsanov theorem framing the problem as a Stochastic Optimal Control (SOC) Problem [Chen et al., 2016, 2021].

$$\min_{u_t, p_t} \int_0^1 \mathbb{E}_{p_t}[\|u_t\|^2]dt \quad \text{s.t.} \ \frac{\partial p_t}{\partial t} = -\nabla \cdot (u_t p_t) + \frac{\sigma^2}{2}\Delta p_t, \quad \text{and} \quad p_0 = \pi_0, \quad p_1 = \pi_1 \tag{82}$$

Finally, note that the static SB is equivalent to the entropy regularized OT formulation [Pavon et al., 2021, Nutz, 2021, Cuturi, 2013].

$$\min_{\pi \in \Pi(\pi_0, \pi_1)} \int_{\mathbb{R}^d \times \mathbb{R}^d} \|x_0 - x_1\|^2 d\pi(x_0, x_1) + \epsilon \mathrm{KL}(\pi|\pi_0 \otimes \pi_1) \tag{83}$$

This regularization term enabled efficient solution through the Sinkhorn algorithm and has presented numerous benefits, such as smoothness, and other statistical properties [Ghosal et al., 2022, Léger, 2021, Peyré et al., 2019].

**Bridge Matching**   Peluchetti [2023] first proposed the Markovian projection to propose Bridge matching, while Liu et al. [2022a] employed it to learn representations in constrained domains. The Bridge matching objective offers a computationally efficient alternative but requires additional assumptions. To this front, Action Matching Neklyudov et al. [2023] presents a general matching method with the least assumptions, at the expense of being unfavorable to scalability. Additionally, recent advances have introduced more general frameworks for conditional generative modeling. Denoising Diffusion Bridge Models (DDBMs) extend traditional diffusion models to handle arbitrary source and target distributions by learning the score of a diffusion bridge, thereby unifying and generalizing methods such as score-based diffusion and flow matching [Zhou et al., 2023]. Similarly, the stochastic interpolant framework [Albergo et al., 2023] integrates flow- and diffusion-based approaches by defining continuous-time stochastic processes that interpolate between distributions. These interpolants achieve exact bridging in finite time by introducing an auxiliary latent variable, offering flexible control over the interpolation path.

Recently, these matching frameworks have been employed to solve the SB problem. DSBM ([Shi et al., 2023]) employs Iterative Markovian Fitting (IMF) to obtain the Schrodinger Bridge solution, while De Bortoli et al. [2023] explores flow and bridge matching processes, proposing a modification to preserve coupling information, demonstrating efficiency in learning mixtures of image translation tasks. $SF^2 - M$[Tong et al., 2023a] provides a simulation-free objective for inferring stochastic dynamics, demonstrating efficiency in solving Schrödinger bridge problems. GSBM Liu et al. [2024] presents a framework for solving distribution matching to account for task-specific state costs. While these methods aim to identify the optimal coupling, [Somnath et al., 2023] and [Liu et al., 2023] propose Bridge Matching algorithms between a priori coupled data, namely the pairing between clean and corrupted images or pairs of biological data from the static Schrödinger Bridge. Lastly, works that aim to improve the efficiency of these matching frameworks by introducing a light solver for implementing optimal matching using Gaussian mixture parameterization [Gushchin et al., 2024a].

**Flow Matching**    In parallel, there have been methodologies that employ deterministic dynamics. Lipman et al. [2022] introduces the deterministic counterpart of Bridge Matching; Flow Matching (FM) for training Continuous Normalizing Flows (CNFs;Chen et al. [2018]) using fixed conditional probability paths. Further developments include Conditional Flow Matching (CFM) offers a stable regression objective for training CNFs without requiring Gaussian source distributions or density evaluations [Tong et al., 2023b], Metric Flow Matching (MFM), which learns approximate geodesics on data manifolds [Kapusniak et al., 2024], and Flow Matching in Latent Space, which improves computational efficiency for high-resolution image synthesis [Dao et al., 2023]. Finally, CFM retrieves exactly the first iteration of the Rectified Flow Liu et al. [2022b], which is an iterative approach for learning ODE models to transport between distributions.

**Multi-Marginal**    Among the advancements of Flow matching models was the introduction of the Multi-Marginal Flow matching framework [Rohbeck et al., 2024]. Similarly to our approach, they proposed a simulation-free training approach, leverages cubic spline-based flow interpolation and classifier-free guidance across time and conditions. TrajectoryNet [Tong et al., 2020] and MIOFlow [Huguet et al., 2022] combine Optimal Transport with Continuous Normalizing Flows [Chen et al., 2018] and manifold embeddings, respectively, to model non-linear continuous trajectories through multiple points. In the stochastic realm, recent works have proposed the mmSB training through extending Iterative Proportional Fitting - a continuous extension of the Sinkhorn algorithm to solve the dynamic SB problem [De Bortoli et al., 2021]- to phase space and adapting the Bregman iterations [Chen et al., 2023a]. Another approach alternates between learning piecewise SBs on the unobserved trajectories and refining the best guess for the dynamics within the specified reference class [Shen et al., 2024]. More recently, modeling the reference dynamics as a special class of smooth Gaussian paths was shown to achieve more regular and interpretable trajectories [Hong et al., 2025]. Furthermore, the multi-marginal problem has been recently addressed by Deep Momentum Multi-Marginal Schrödinger Bridge [DMSB;Chen et al. [2023a]] proposed to solve the mmSB in phase space via adapting the Bregman iterations. More recently, an iterative method for solving the mmSB proposed learning piecewise SB dynamics within a preselected reference class [Shen et al., 2024]. In contrast, modeling the reference dynamics as smooth Gaussian paths was shown to achieve more temporally coherent and smooth trajectories [Hong et al., 2025], though the belief propagation prohibits scaling in high dimensions. Lastly, Wasserstein Lane–Riesenfeld (WLR) is a geometry-aware method to reconstruct smooth trajectories from point clouds. It performs consecutive geodesic averaging in Wasserstein space, giving spline-like curves that can handle mass splitting/bifurcations, achieving strong results on cell datasets [Banerjee et al., 2024].

# E    Additional Details on Experiments

## E.1    General Information

In this section, we revisit our experimental results to evaluate the performance of 3MSBM on a variety of trajectory inference tasks, such as Lotka-Volterra, ocean current in the Gulf of Mexico, single-cell sequencing, and the Beijing air quality data. We compared against state-of-the-art methods explicitly designed to incorporate multi-marginal settings, such Deep Momentum Multi-Marginal Schrödinger Bridge (DMSB;Chen et al. [2023a]), Schrodinger Bridge with Iterative Reference Refinement (SBIRR; Shen et al. [2024]), smooth Schrodinger Bridges (SmoothSB; Hong et al. [2025]), and Multi-Marginal Flow Matching (MMFM; Rohbeck et al. [2024]), along with two NeuralODE-based methods: MIOFlow [Huguet et al., 2022] and DeepRUOT [Zhang et al., 2024b]. We used the official implementations of all compared methods, with default hyperparameters. For all experiments with our 3MSBM, we employed the ResNet architectures from Chen et al. [2023b], Dockhorn et al. [2021]. We used the AdamW optimizer and applied Exponential Moving Averaging with a decay rate of 0.999. All results are averaged over 5 random seeds, with means and standard deviations reported in Section 4 and the tables below. Experiments were run on an RTX 4090 GPU with 24 GB of VRAM.

## E.2    Lotka-Volterra

We first consider a synthetic dataset generated by the Lotka–Volterra (LV) equations [Goel et al., 1971], which model predator-prey interactions through coupled nonlinear dynamics. We used the dataset from [Shen et al., 2024] with the 5 training and 4 validation time points, with 50 observations

Table 9: Mean of SWD, MMD, $\mathcal{W}_1$, and $\mathcal{W}_2$ over 5 seeds at left-out marginals on the LV dataset for DeepRUOT, MIOFlow, SBIRR, MMFM, Smooth SB, and 3MSBM (Lower is better).

(a) SWD

| Method | $t_1$ | $t_3$ | $t_5$ | $t_7$ |
|---|---|---|---|---|
| DeepRUOT | 0.30 | 0.16 | 0.22 | 0.44 |
| SBIRR | 0.17 | 0.18 | 0.24 | 0.48 |
| MIOFlow | 1.53 | 1.49 | 1.23 | 1.46 |
| Smooth SB | 0.49 | 0.18 | 0.13 | 0.37 |
| DMSB | 0.64 | 0.67 | 0.98 | 0.63 |
| MMFM | 0.21 | 0.25 | 0.39 | 0.57 |
| 3MSBM | 0.29 | 0.13 | 0.11 | 0.37 |

(b) MMD

| Method | $t_1$ | $t_3$ | $t_5$ | $t_7$ |
|---|---|---|---|---|
| DeepRUOT | 3.02 | 0.20 | 0.39 | 0.48 |
| SBIRR | 0.44 | 0.53 | 0.46 | 0.50 |
| MIOFlow | 7.09 | 6.52 | 6.29 | 5.28 |
| Smooth SB | 3.23 | 1.21 | 0.32 | 0.42 |
| DMSB | 3.46 | 3.43 | 1.16 | 1.74 |
| MMFM | 0.52 | 0.60 | 0.63 | 0.77 |
| 3MSBM | 4.33 | 0.72 | 0.40 | 0.40 |

(c) $\mathcal{W}_1$

| Method | $t_1$ | $t_3$ | $t_5$ | $t_7$ |
|---|---|---|---|---|
| DeepRUOT | 0.40 | 0.17 | 0.29 | 0.62 |
| MIOFlow | 2.53 | 2.12 | 1.76 | 1.53 |
| SBIRR | 0.19 | 0.20 | 0.30 | 0.48 |
| Smooth SB | 0.29 | 0.27 | 0.22 | 0.68 |
| DMSB | 0.99 | 0.74 | 0.60 | 0.98 |
| MMFM | 0.24 | 0.43 | 0.57 | 0.75 |
| 3MSBM | 0.23 | 0.18 | 0.12 | 0.35 |

(d) $\mathcal{W}_2$

| Method | $t_1$ | $t_3$ | $t_5$ | $t_7$ |
|---|---|---|---|---|
| DeepRUOT | 0.44 | 0.19 | 0.31 | 0.65 |
| MIOFlow | 2.77 | 2.34 | 1.76 | 1.55 |
| SBIRR | 0.19 | 0.25 | 0.45 | 0.74 |
| Smooth SB | 0.27 | 0.27 | 0.22 | 0.68 |
| DMSB | 0.84 | 0.98 | 0.71 | 1.24 |
| MMFM | 0.22 | 0.43 | 0.77 | 1.23 |
| 3MSBM | 0.24 | 0.17 | 0.15 | 0.36 |

per time point. In particular, the generated dataset consists of 9 marginals in total; the even-numbered indices are used to train the model (i.e., $t_0, t_2, t_4, t_6, t_8$), and the remainder of the time points are used to assess the efficacy of our model to impute and infer the missing time points. In this experiment, we benchmarked 3MSBM against DeepRUOT, MIOFlow, DMSB, SBIRR, MMFM, and Smooth SB. Table 9 reports the mean performance over 5 seeds of each method with respect to the SWD, MMD, $\mathcal{W}_1$, and $\mathcal{W}_2$ distances from the validation points. The hyperparameter selection for the LV with our method were: the diffusion coefficient was set to $\sigma = 0.3$, and the learning rate was $10^{-4}$.

## E.3 Gulf of Mexico

Subsequently, we evaluate the efficacy of our model to infer the missing time points in a real-world multi-marginal dataset. The dataset contains ocean-current snapshots of the velocity field around a vortex in the Gulf of Mexico (GoM). Similarly to the LV dataset, we used the big vortex dataset provided in [Shen et al., 2024], consisting of 300 samples across 5 training and 4 validation times. More explicitly, out of the total 9 marginals, the even-indexed time points (i.e., $t_0, t_2, t_4, t_6, t_8$) are used for training, and the remaining are left out to evaluate the model's ability to impute and infer missing temporal states. We compared our 3MSBM against DeepRUOT, MIOFlow, DMSB, SBIRR, MMFM, and Smooth SB for this experiment. Table 10a demonstrates the mean performance over 5 seeds of each method with respect to the SWD, MMD, $\mathcal{W}_1$, and $\mathcal{W}_2$ distances from the validation points. The hyperparameters used for the GoM experiment with our method were: a batch size of 32 for the matching, the diffusion coefficient was set to $\sigma = 0.3$, and the learning rate was set equal to $2 \cdot 10^{-4}$.

## E.4 Beijing air quality

We revisit our experiments using the Beijing multi-site air quality data set [Chen, 2017]. This dataset consists of hourly air pollutant data from 12 air-quality monitoring sites across Beijing. We focus on PM2.5 data, an indicator monitoring the density of particles smaller than 2.5 micrometers, from January 2013 to January 2015. We focused on a single monitoring site and aggregated the measurements collected within the same month. To introduce temporal separation between observations, we selected measurements from every other month, resulting in 13 temporal snapshots. For the imputation task, we omitted the data at $t_2, t_5, t_8$, and $t_{11}$, while the remaining snapshots formed the training set. Table 11a shows the mean performance over 5 seeds of each method in the SWD, MMD, $\mathcal{W}_1$, and $\mathcal{W}_2$ distances from the validation time points, benchmarking our 3MSBM method against

Table 10: Mean of SWD, MMD, $\mathcal{W}_1$, and $\mathcal{W}_2$ over 5 seeds at left-out marginals on the GoM dataset for DeepRUOT, MIOFlow, SBIRR, MMFM, Smooth SB, and 3MSBM (Lower is better).

(a) SWD

| Method | $t_1$ | $t_3$ | $t_5$ | $t_7$ |
|---|---|---|---|---|
| DeepRUOT | 0.21 | 0.33 | 0.23 | 0.21 |
| MIOFlow | 0.83 | 0.34 | 1.23 | 0.97 |
| SBIRR | 0.15 | 0.11 | 0.11 | 0.09 |
| Smooth SB | 0.17 | 0.14 | 0.10 | 0.13 |
| DMSB | 0.23 | 0.54 | 0.39 | 0.28 |
| MMFM | 0.23 | 0.25 | 0.10 | 0.19 |
| 3MSBM | 0.14 | 0.14 | 0.08 | 0.06 |

(b) MMD

| Method | $t_1$ | $t_3$ | $t_5$ | $t_7$ |
|---|---|---|---|---|
| DeepRUOT | 2.35 | 1.75 | 1.30 | 1.10 |
| MIOFlow | 6.52 | 4.13 | 6.88 | 6.11 |
| SBIRR | 4.65 | 0.82 | 1.15 | 0.35 |
| Smooth SB | 4.98 | 4.02 | 0.84 | 0.28 |
| DMSB | 4.50 | 4.40 | 4.57 | 3.37 |
| MMFM | 0.91 | 1.20 | 0.49 | 0.51 |
| 3MSBM | 3.99 | 2.92 | 0.87 | 0.52 |

(c) $\mathcal{W}_1$

| Method | $t_1$ | $t_3$ | $t_5$ | $t_7$ |
|---|---|---|---|---|
| DeepRUOT | 0.29 | 0.29 | 0.20 | 0.25 |
| MIOFlow | 1.10 | 0.55 | 1.67 | 1.69 |
| SBIRR | 0.28 | 0.15 | 0.11 | 0.15 |
| Smooth SB | 0.16 | 0.27 | 0.21 | 0.56 |
| DMSB | 0.24 | 0.31 | 0.70 | 0.46 |
| MMFM | 0.33 | 0.38 | 0.22 | 0.29 |
| 3MSBM | 0.17 | 0.21 | 0.09 | 0.12 |

(d) $\mathcal{W}_2$

| Method | $t_1$ | $t_3$ | $t_5$ | $t_7$ |
|---|---|---|---|---|
| DeepRUOT | 0.32 | 0.43 | 0.36 | 0.33 |
| MIOFlow | 1.11 | 0.48 | 1.68 | 1.70 |
| SBIRR | 0.24 | 0.13 | 0.21 | 0.17 |
| Smooth SB | 0.22 | 0.27 | 0.21 | 0.16 |
| DMSB | 0.28 | 0.30 | 0.72 | 0.46 |
| MMFM | 0.33 | 0.32 | 0.19 | 0.31 |
| 3MSBM | 0.20 | 0.18 | 0.07 | 0.09 |

Table 11: Mean of SWD, MMD, $\mathcal{W}_1$, and $\mathcal{W}_2$ over 5 seeds at left-out marginals on the Beijing Air Quality dataset for DeepRUOT, MIOFlow, MMFM, Smooth SB, and 3MSBM (Lower is better).

(a) SWD

| Method | $t_1$ | $t_3$ | $t_5$ | $t_7$ |
|---|---|---|---|---|
| DeepRUOT | 13.67 | 52.60 | 71.34 | 84.67 |
| MIOFlow | 46.64 | 79.06 | 76.06 | 60.87 |
| Smooth SB | 28.61 | 28.81 | 35.79 | 32.90 |
| DMSB | 21.10 | 21.92 | 35.53 | 35.75 |
| MMFM | 17.51 | 23.94 | 32.56 | 39.98 |
| 3MSBM | 17.70 | 9.78 | 22.23 | 32.23 |

(b) MMD

| Method | $t_1$ | $t_3$ | $t_5$ | $t_7$ |
|---|---|---|---|---|
| DeepRUOT | 0.36 | 0.34 | 0.99 | 1.67 |
| MIOFlow | 0.58 | 0.92 | 0.38 | 0.51 |
| Smooth SB | 0.41 | 0.43 | 0.39 | 0.48 |
| DMSB | 0.76 | 0.79 | 0.54 | 0.47 |
| MMFM | 0.44 | 0.56 | 0.59 | 0.55 |
| 3MSBM | 0.35 | 0.85 | 0.28 | 0.32 |

(c) $\mathcal{W}_1$

| Method | $t_1$ | $t_3$ | $t_5$ | $t_7$ |
|---|---|---|---|---|
| DeepRUOT | 10.49 | 39.82 | 51.00 | 68.97 |
| MIOFlow | 31.79 | 56.35 | 57.89 | 45.36 |
| Smooth SB | 23.73 | 23.88 | 24.70 | 28.61 |
| DMSB | 58.79 | 32.70 | 40.22 | 42.06 |
| MMFM | 28.08 | 26.40 | 37.73 | 51.12 |
| 3MSBM | 12.71 | 57.44 | 26.02 | 29.61 |

(d) $\mathcal{W}_2$

| Method | $t_1$ | $t_3$ | $t_5$ | $t_7$ |
|---|---|---|---|---|
| DeepRUOT | 13.67 | 52.59 | 71.35 | 84.67 |
| MIOFlow | 46.64 | 79.06 | 76.06 | 60.87 |
| Smooth SB | 28.61 | 28.81 | 35.79 | 32.90 |
| DMSB | 60.19 | 38.77 | 41.38 | 43.25 |
| MMFM | 26.42 | 29.95 | 43.49 | 49.96 |
| 3MSBM | 12.87 | 79.36 | 27.89 | 32.26 |

MMFM with cubic splines, DeepRUOT, MIOFlow, and Smooth SB. Note, for this experiment, we did not benchmark against SBIRR, since we did not possess the corresponding informative prior measure. The hyperparameters used for the Beijing air quality experiment with our method were: a total number of samples of 1000 were used, with a batch size of 64 for the matching, the diffusion coefficient was set to $\sigma = 0.2$, and the learning rate was set to $5 \cdot 10^{-5}$.

## E.5 Single sequencing

Lastly, we revisit our experiments on the Embryoid Body (EB) stem cell differentiation dataset, which captures cell progression across 5 stages over a 27-day period. Following the setup in Section 4.4, we used the preprocessed data from [Tong et al., 2020, Moon et al., 2019], embedded

in a 100-dimensional PCA feature space. Cell snapshots were collected at five discrete intervals: $t_0 \in [0,3]$, $t_1 \in [6,9]$, $t_2 \in [12,15]$, $t_3 \in [18,21]$, $t_4 \in [24,27]$. Below, we present the results of the comparison of our 3MSBM against DeepRUOT, MIOFlow, DMSB, SBIRR, MMFM. Table 10a demonstrates the mean performance over 5 seeds of each method with respect to the SWD, MMD, $\mathcal{W}_1$, and $\mathcal{W}_2$ distances from the validation points, i.e., at the snapshots $t_1$, and $t_3$. Observing the results in Table 12 is evident that our 3MSBM consistently outperforms the state-of-the-art algorithms in the high-dimensional EB-100 task across all metrics. The hyperparameters used for every EB experiment with our method were: a total number of samples of 1000 were used, with a batch size of 64 for the matching, the diffusion coefficient was set to $\sigma = 0.1$, and the learning rate was set to $10^{-4}$.

Table 12: Mean of SWD, MMD, $\mathcal{W}_1$, and $\mathcal{W}_2$ over 5 seeds at left-out marginals on the EB-100 data for DeepRUOT, MIOFlow, SBIRR, MMFM, Smooth SB, and 3MSBM (Lower is better).



(a) SWD

| Method | $t_1$ | $t_3$ |
|---|---|---|
| DeepRUOT | 0.73 | 0.67 |
| MIOFlow | 0.84 | 0.94 |
| SBIRR | 0.80 | 0.91 |
| DMSB | 0.58 | 0.54 |
| MMFM | 0.59 | 0.76 |
| 3MSBM | 0.48 | 0.38 |

(b) MMD

| Method | $t_1$ | $t_3$ |
|---|---|---|
| DeepRUOT | 0.43 | 0.36 |
| MIOFlow | 1.01 | 0.92 |
| SBIRR | 0.71 | 0.73 |
| DMSB | 0.38 | 0.36 |
| MMFM | 0.37 | 0.35 |
| 3MSBM | 0.18 | 0.14 |

(c) $\mathcal{W}_1$

| Method | $t_1$ | $t_3$ |
|---|---|---|
| DeepRUOT | 13.45 | 14.90 |
| MIOFlow | 13.20 | 13.57 |
| SBIRR | 15.09 | 20.39 |
| DMSB | 14.08 | 15.22 |
| MMFM | 13.61 | 14.64 |
| 3MSBM | 13.89 | 13.11 |

(d) $\mathcal{W}_2$

| Method | $t_1$ | $t_3$ |
|---|---|---|
| DeepRUOT | 13.64 | 15.10 |
| MIOFlow | 13.66 | 14.05 |
| SBIRR | 15.42 | 20.98 |
| DMSB | 14.83 | 15.49 |
| MMFM | 14.68 | 14.83 |
| 3MSBM | 14.51 | 13.26 |



### E.6 Ablation study on $\sigma$

In stochastic optimal control [Theodorou et al., 2010], the value of $\sigma$ plays a crucial role in representing the uncertainty from the environment or the error in applying the control. As a result, the optimal control policy can vary significantly with different degrees of noise. Figure 14 demonstrates the performance of our 3MSBM with respect to varying noise in the EB and GoM datasets. We observe consistent performance in the training marginals across all tested values of $\sigma$, whereas for the points in the validation set, increasing $\sigma$ up to a point is deemed beneficial as it improves performance. Sample trajectories in GoM in Figure 13 further verify this trend. Low noise values (e.g. $\sigma = 0.05$) cause the trajectories to be overly tight, whereas at high noise (e.g. $\sigma = 1.0$), the trajectories become overly diffuse. On the other hand, moderate noise values (e.g. $\sigma = 0.4$) achieve a good balance, enabling well-spread trajectories matching the validation marginals.

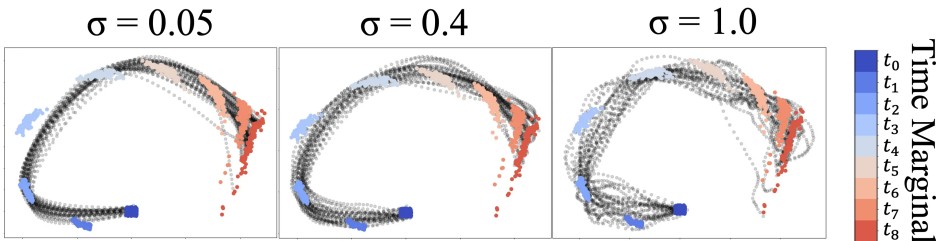

Figure 13: Comparison of the trajectories inferred on the Gulf Mexico current dataset for different values of $\sigma$

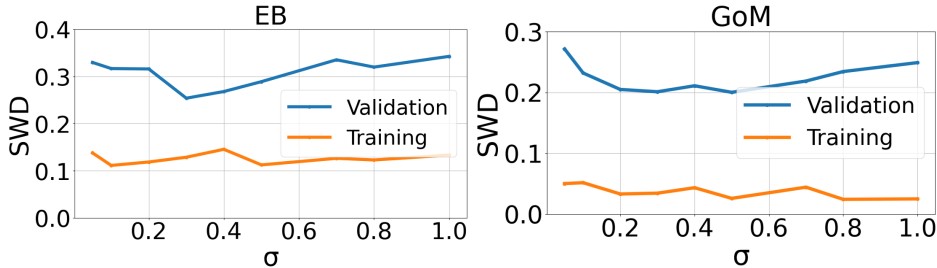

Figure 14: SWD from the marginals in the Validation and Training set for varying values of sigma on EB and GoM

