# OpenReview forum: "Momentum Multi-Marginal Schrödinger Bridge Matching"
_NeurIPS.cc/2025/Conference — NeurIPS 2025 poster_

### Official Review · Reviewer_wHqP · 2025-06-27

**Clarity:** 3
**Significance:** 3
**Originality:** 3
**Rating:** 5
**Confidence:** 4

**Summary:**

This paper proposes 3MBSM, a method for performing distributionally optimal interpolation based on multiple snapshot data to infer the continuous-time dynamics of a system. The authors address the Momentum Multi-Marginal Schrödinger Bridge problem (a type of Schrödinger Bridge formulated in phase space), which compensates for the shortcomings of existing methods in capturing long-term temporal dependencies and ensuring trajectory continuity by simultaneously considering the coupling among multiple marginal distributions and incorporating both the position and momentum information of the system. Initially, the paper derives an analytical solution for the conditional acceleration function \(a(t)\) using an optimal control approach, and then employs neural networks to approximate the unconditional \(a(t)\). The proposed method is evaluated on multiple datasets, and the experimental results demonstrate that its error metrics for inferring trajectories at missing time points are superior to those of mainstream methods.

**Questions:**

Based on the above weaknesses, I have the following questions:

1. The method in the paper estimates the initial velocity distribution through forward and backward iterations, with each iteration requiring numerical integration of the SDE. Would multiple numerical integrations significantly increase the computational complexity of the algorithm and affect its scalability? Could this issue be discussed?

2. The efficiency in sampling of the proposed method stems from the fact that if the conditional probability path is Gaussian, the mean and covariance of the Gaussian distribution have analytical solutions. This arises because \(a(t)\) depends linearly on \(x(t)\) and \(v(t)\) (in essence, this again comes from the second-order approximation of the value function). Is this linearization of the system too crude? Can it handle systems with more complicated dynamics and strong nonlinearities? Are there any potential solutions to this problem?

3. Could more experiments be included? For example, on the current datasets, it would be valuable to compare the proposed method with all baselines using a unified evaluation metric; perform quantitative evaluations on the single-cell dataset; and compare the proposed method with Neural-ODE approaches such as Mioflow [1] and DeepRUOT [2].

4. What is the rationale behind selecting a second-order dynamical system in the paper? Using a second-order dynamical system for interpolation to obtain smoother trajectories appears to be a trivial conclusion, similar to how higher-order polynomial interpolation yields smoother trajectories. Could the authors discuss what additional information about the system dynamics is captured by using a second-order dynamical system for interpolation?

5. Could a comparison be made between the proposed method and the method in DMSB (since both papers tackle the same problem)? Specifically, how much computational overhead is reduced by the current method compared to DMSB? From an optimal control perspective, which method is superior? (For instance, is there a significant difference in the cost functions of the trajectories obtained by the two methods, and which one yields lower cost?)

Overall, I provide a positive and optimistic evaluation of this work. If the authors could address the issues raised, with additional experiments or further discussion, I would be very willing to raise my score further.

[1] Manifold Interpolating Optimal-Transport Flows for Trajectory Inference, Guillaume Huguet, D.S. Magruder, Alexander Tong et al., NeurIPS 2022.

[2] Learning stochastic dynamics from snapshots through regularized unbalanced optimal transport, Zhenyi Zhang, Tiejun Li, Peijie Zhou, ICLR 2025.

**Ethical Concerns:**

["NO or VERY MINOR ethics concerns only"]

**Final Justification:**

The author's reply has shown me the great potential of this method in learning complex dynamics with stochastic effect, addressing all my concerns. Assuming all these things are incorporated in the revised version, I am very happy to raise my score to 5.

**Limitations:**

While this study provides valuable insights, a key limitation lies in the limited consideration of unnormalized distributions in the limitations section.

**Quality:**

3

**Strengths And Weaknesses:**

**Strengths:**

1. The paper is generally well-written and presents extensive experiments on multiple datasets, overall offering a comprehensive study.

2. The theoretical analysis in the paper is sufficiently thorough. By employing a second-order approximation of the value function and using optimal control methods, the paper derives the differential equation governing the value function and the corresponding optimal control law, as well as the analytical solution for \(a(t)\). Additionally, the paper proves that the proposed iterative framework has a unique fixed point.

3. The idea presented in the paper is very interesting. In the field of trajectory inference, most current work is based on a first-order dynamical system framework; using a second-order dynamical system to describe the underlying dynamics in the data is a promising and worthy direction for exploration.


**Weaknesses:**

1. The paper employs a training strategy similar to Flow Matching: first, a conditional probability path is obtained by solving an optimal control problem, and then a neural network is used to regress the unconditional \(a(t)\). However, before each update of \(a(t)\) in the training algorithm, iterations are required to estimate the initial distribution of \(v(t)\), and each iteration involves numerical integration of an SDE. This limits the scalability of the algorithm.

2. The paper effectively linearizes the system, making the control law \(a(t)\) linearly dependent on \(x(t)\) and \(v(t)\), which restricts the solution accuracy of the algorithm.

3. The experimental evaluation in the paper is not sufficiently comprehensive. Although the paper plans to compare the proposed 3MBSM method with SBIRR, DMSB, SmoothSB, and MMFM, only a subset of these methods is compared on each dataset (for example, only MMFM and DMSB are compared on the Beijing Air Quality dataset). Additionally, the evaluation metrics vary across datasets (e.g., the Gulf of Mexico dataset uses the W1 and W2 distances, while other datasets employ SWD and MMD). In the high-dimensional single-cell dataset, only qualitative evaluations are provided, with no quantitative assessments.

4. The motivation for using a second-order dynamical system in the paper is unclear; the justification of "yielding smoother trajectories" does not appear to be a decisive factor for adopting a second-order system. Furthermore, employing a second-order dynamical system introduces additional complexity, potentially making the fitting process more challenging.

5. The paper does not evaluate the method from an optimal control perspective, for instance, by comparing the cost functions of the trajectories with those obtained by other methods.

6. The papre dose not consider the unbalancedness of the dataset, which is very important in the single cell dynamics.

---

> ### Author Rebuttal · Authors · 2025-07-31
>
> We would like to thank the reviewer for their positive remarks, as well as their useful comments and further suggestions for improving our work. In the following, we address each of the concerns and suggestions of the reviewer.
>
> ### **1. Computational complexity of solving 3MBB**
>
> We highlight that the step of iteratively propagating the conditional dynamics—to approximate the initial velocity— introduces minimal computational overhead. Below, we empirically validate this claim by presenting a breakdown of the time percentage attributed to each of the two steps of our algorithm: i) the iterative propagation of the conditional dynamics (*lines 4-9 in Alg. 1)* and ii) the Bridge Matching step *(line 10 in Alg. 1)* take in the EB-100 and GoM datasets. We see that the computational complexity introduced by the first step of the algorithm, i.e., the iterative propagation of the conditional dynamics, remains negligible compared to the Bridge Matching.
>
> | Dataset | Cond. Bridge | Bridge Matching |
> | --- | --- | --- |
> | EB-100 | 3.3 % | 96.7 % |
> |    GoM  | 7.4 % | 92.6 % |
>
> This efficiency stems from two key factors: i) convergence to the initial velocity is typically achieved within only ~10 iterations, ii) most importantly, the backward ODEs admit analytic solutions, allowing us to compute the conditional acceleration in closed form. This eliminates the need to perform costly numerical integration to solve Eq. (5), which would cause computational bottlenecks and error accumulation.
>
> ### **2. Discussion on linearization**
>
> We would like to thank the reviewer for this important and insightful question.
>
> We emphasize that the linear expression of the conditional $a_t$ in Eq. (6) is not an approximation but the exact solution to the optimization problem in Eqs. (3) & (4). This comes from the fact that its objective (Eq. (3)) is quadratic in the augmented state $m_t$, and that its dynamics (Eq. (4)) are control-affine with additive noises. In such cases, the optimization is of a standard Linear-Quadratic Regulator (LQR) form, where the value function is known to be quadratic [3]. That said, the quadratic form of value function in L142 and its backward ODEs in Eq 5 are exact—rather than approximated, and, consequently, the optimal conditional $a_t$ is linear in $(x_t,v_t)$. We stress this distinction, but also acknowledge that current writing may confuse readers. We’ll update this part in the revision, and we thank the reviewer for raising the comments.
>
> Regarding the expression of Eq. (3), which relaxes the original interpolation problem (i.e., solving Eq. (2) with $m_n$ pinned at $\bar m_n$ for all $n \in N$) to a soft quadratic penalty. We note that this relaxation follows prior frameworks [1,2]—which transforms the problem into a multi-marginal conditional stochastic optimal control problem—and the soft penalty can be made hard by taking the penalty matrix $R$ to infinity, as illustrated in Prop 3.2.
>
> ### **3. Second-order dynamics**
>
> To capture coherent long-term dependencies in sparse, irregularly sampled time series, we incorporate second-order dynamics into our multi-point inference framework. By minimizing path acceleration, the model achieves low-curvature, smooth and consistent trajectories. Velocity provides a memory, enabling the system to preserve trends such as acceleration or deceleration that are lost in first-order (memoryless) models. This enables the capture of nonlinear transitions and inflection points, leading to more realistic interpolations that align with the behavior of physical systems—many of which (e.g., Newtonian or Hamiltonian systems) are governed by second-order dynamics. This is particularly important when observations are coarse or widely spaced. In contrast, first-order models often produce unnatural kinks or discontinuities (see Fig. 1).
>
> Our approach overcomes these challenges by solving the momentum multi-marginal Schrödinger Bridge problem, jointly optimizing across all timepoints. This yields globally optimal couplings and enables physically plausible long-range inference, as evidenced by the superior performance in Section 4.
>
> ### **4. Additional Experiments**
>
> We thank the reviewer for suggesting a more comprehensive experimental comparison. Below, we present additional results on the same datasets used in the main paper (GoM, LV, EB-100, and Beijing Air Quality), following the original experimental setups. Alongside the trajectory inference baselines included in the paper (i.e, SBIRR [4], Smooth SB [5], DMSB [6], and MMFM [7]) we also evaluate against two NeuralODE-based methods: MIOFlow [8] and DeepRUOT [9]. We use the **same metric** in all datasets: the **Sliced Wasserstein Distance (SWD)**.
>
> Note that the Smooth SB could not scale to the 100-dimensional single-cell dataset (EB-100), and thus, results are not available for this setting. Our 3MSBM consistently achieves the lowest or near-lowest SWD across all datasets and timepoints, demonstrating its ability to generate smooth, temporally coherent interpolations even in the presence of complex, multi-modal dynamics. Especially, on the single-cell EB-100 dataset, 3MSBM shows strong imputation capabilities, accurately interpolating both left-out marginals and achieving the lowest SWD scores.
> - **SWD on GoM**
> | Method | $t_1$ | $t_3$ | $t_5$ | $t_7$ |
> | --- | --- | --- | --- | --- |
> | DeepRUOT | 0.21 | 0.33 | 0.23 | 0.21 |
> | MIOFlow | 0.83 | 0.34 | 1.23 | 0.97 |
> | SBIRR | 0.15 | **0.11** | 0.11 | 0.09 |
> | Smooth SB | 0.17 | 0.14 | 0.10 | 0.13 |
> | DMSB | 0.23 | 0.54 | 0.39 | 0.28 |
> | MMFM | 0.23 | 0.25 | 0.10 | 0.19 |
> | 3MSBM | **0.14** | 0.14 | **0.08** | **0.06** |
>
> - **SWD on LV**
> | Method | $t_1$ | $t_3$ | $t_5$ | $t_7$ |
> | --- | --- | --- | --- | --- |
> | DeepRUOT | 0.30 | 0.16 | 0.22 | 0.44 |
> | MIOFlow | 1.53 | 1.49 | 1.23 | 1.46 |
> | SBIRR | **0.17** | 0.18 | 0.24 | 0.48 |
> | Smooth SB | 0.49 | 0.18 | 0.13 | 0.37 |
> | DMSB | 0.64 | 0.67 | 0.98 | 0.63 |
> | MMFM | 0.21 | 0.25 | 0.39 | 0.57 |
> | 3MSBM | 0.29 | **0.13** | **0.11** | **0.37** |
>
> - **SWD on EB-100**
> | Method | $t_1$ | $t_3$ |
> | --- | --- | --- |
> | DeepRUOT | 0.73 | 0.67 |
> | MIOFlow | 0.84 | 0.94 |
> | SBIRR | 0.80 | 0.91 |
> | Smooth SB | -- | -- |
> | DMSB | 0.58 | 0.54 |
> | MMFM | 0.59 | 0.76 |
> | 3MSBM | **0.48** | **0.38** |
>
> - **SWD on BAQ**
> | Method | $t_2$ | $t_5$ | $t_8$ | $t_{11}$ |
> | --- | --- | --- | --- | --- |
> | DeepRUOT | **13.67** | 52.60 | 71.34 | 84.67 |
> | MIOFlow | 46.64 | 79.06 | 76.06 | 60.87 |
> | SBIRR | 47.84 | 60.40 | 55.03 | 59.12 |
> | Smooth SB | 28.61 | 28.81 | 35.79 | 32.90 |
> | DMSB | 51.10 | 37.92 | 40.53 | 35.75 |
> | MMFM | 17.51 | 23.94 | 32.56 | 39.98 |
> | 3MSBM | 17.70 | **9.78** | **22.23** | **32.23** |
>
> ### **5. Comparisons between 3MSBM and DMSB**
>
> Indeed, our 3MSBM as a matching framework provides enhanced scalability compared to prior IPF methods solving the mmSB problem. To empirically verify our claims:
>
> 1. We compare the total training time of our algorithm to DMSB on all datasets and observe that 3MSBM is consistently and significantly faster in wall-clock time across all datasets.
>
>     | Time (hours:min) | EB-100 | GoM | LV | BAQ |
>     | --- | --- | --- | --- | --- |
>     | DMSB | 5:51 | 2:55 | 2:43 | 3:15 |
>     | 3MSBM | 2:18 | 0:34 | 0:21 | 2:03 |
>
> 2. We ablate the batch size in the EB-100 dataset and report the peak memory usage compared to DMSB. We find that our method consistently requires less memory than the IPF-based DMSB, which encounters a memory error for larger batch sizes. This enables significantly larger batch sizes, even in high-dimensional settings, providing greater design flexibility without compromising performance.
>
>
>     | Batch Size | DMSB | 3MSBM |
>     | --- | --- | --- |
>     | 256 | 6.1 GB | 1.8 GB |
>     | 512 | 12.3 GB | 3.3 GB |
>     | 1024 | -- | 6.7 GB |
>     | 2048 | -- | 13.8 GB |
>
> Additionally, we evaluate our method from an optimal control perspective by computing the objective $\int_0^1 \|a_t\|^2dt$ . We compare the performance of our 3MSBM with DMSB in terms of this control cost, which directly reflects the mmSB objective. Across all experiments, 3MSBM consistently achieves lower control cost, indicating greater optimality, while satisfying the marginal constraints more accurately, as shown in the experiments above.
>
> | $\int_0^1 \|a_t\|^2dt$ | EB-100 | GoM | LV | BAQ |
> | --- | --- | --- | --- | --- |
> | DMSB | 2347.32 | 31.86 | 105.52 | 204.7 |
> | 3MSBM | 1160.96 | 18.45 | 89.75 | 158.9 |
>
> ### **6. Unbalancedness**
>
> While unbalanced dynamics were not within the scope of our current work, we agree with the reviewer on their importance for modeling single-cell systems, and we thank them for this insightful suggestion. We plan to investigate this direction by extending our mmSB objective through the lens of unbalanced OT, in the spirit of the DeepRUOT framework. Specifically, we believe it is possible to incorporate a regularization term in the mmSB objective accounting for cell birth and death rates, analogous to the works in [1, 2]. That said, due to this generalization, it is likely that the analytic solution derived in Proposition 3.2 might not be applicable in this instance. Nevertheless, we agree with the reviewer that this is an intriguing direction for future work.
>
> ---
>
> ### References
>
> 1. Generalized Schrödinger bridge matching
> 2. Feedback Schrödinger Bridge matching
> 3. Stochastic bridges of linear systems
> 4. Multi-marginal Schrödinger Bridges with iterative reference refinement
> 5. Trajectory inference with smooth Schrödinger Bridges
> 6. Deep momentum multi-marginal Schrödinger Bridge
> 7. Modeling complex system dynamics with flow matching across time and conditions.
> 8. Manifold Interpolating Optimal-Transport Flows for Trajectory Inference
> 9. Learning stochastic dynamics from snapshots through regularized unbalanced optimal transport.

---

> > ### Comment · Reviewer_wHqP · 2025-08-04
> >
> > Thank you very much for the thorough response, as well as the additional experiments provided. The author's reply has shown me the great potential of this method in learning complex dynamics with stochastic effect, addressing all my concerns. Assuming all these things are incorporated in the revised version, so I am very happy to raise my score to 5.

---

### Official Review · Reviewer_Pewn · 2025-06-29

**Clarity:** 2
**Significance:** 2
**Originality:** 2
**Rating:** 4
**Confidence:** 2

**Summary:**

The authors present a method for Schrödinger bridge matching via a phase space formulation and for multiple marginals. It alternates between a closed-form momentum bridge and drift fitting, preserving all snapshot constraints. Authors claim convergence and faster training. Experiments on four datasets show lower transport errors than prior methods.

**Questions:**

- What is the capital R in Algorithm 1? Previously this was defined as a matrix.
- How does training cost scale with the number of time points? Do training costs ultimately constrain the batch sizes one can use, how does this in turn affect accuracy?
- The alternating scheme between interpolating paths and coupling refinements feels like it could be brittle for less smooth trajectories or multi-modal marginals. Do the authors have any sense of the limitations of their method in terms of these issues?
- It would be great to see a table (or something like this) clearly showing the differentiate factors and or advantages of their method compared to the many other "multi-marginal" and/or "momentum" Schrödinger Bridge approaches.

**Ethical Concerns:**

["NO or VERY MINOR ethics concerns only"]

**Final Justification:**

The author resolved all my major concerns. I think this paper is a reasonable additional to the conference, but maybe not an extremely strong one.

**Limitations:**

Yes.

**Quality:**

3

**Strengths And Weaknesses:**

**Strengths**
- The paper is well written.
- The motivation of trying to capture long range dependence in trajectory inference type problems is interesting.
- The numerical results feel well founded and generally point to a strong method.

**Weaknessess**
- There are many variants for "multi-marginal" and/or "momentum" Schrödinger Bridge approaches in the literature. The authors do not do a great job of clearly showing the clear differentiate factors and or advantages of their method (aside from improvements on some benchmark problems).
- Scaling claims are somewhat unsubstantiated; how does the ability to training scale with number of samples, dimensional of samples etc? Where do other methods fail at scaling? What experiments demonstrate this? The authors leave high dimensional problems to future work.

---

> ### Author Rebuttal · Authors · 2025-07-31
>
> We thank the reviewer for their positive comments, constructive criticism, and helpful suggestions. Below, we address the raised points.
>
> ### **1. 3MSBM vs prior works**
>
> In the revision, we elaborated on Table 1 and modified the Introduction to better present the differentiating factors of our 3MSBM compared to prior momentum and multi-marginal methods. The table below compares our method against several state-of-the-art multi-marginal approaches tailored for trajectory inference, based on three key attributes:
>
> - Simulation-free training: indicating scalability
> - Globally optimal coupling: assessing whether the method retrieves the mmSB solution versus relying on local approximations.
> - Smooth and coherent trajectories: reflecting second-order dynamics
>
> |  | SBIRR | MMFM | MMSFM | DMSB | Smooth SB | 3MSBM |
> | --- | --- | --- | --- | --- | --- | --- |
> | Simulation Free Training | ☑ | ☑ | ☑ | ☒ | ☒ | ☑ |
> | Globally Optimal Coupling | ☒ | ☒ | ☒ | ☑ | ☑ | ☑ |
> | Smooth Coherent Trajectories | ☒ | ☑ | ☒ | ☑ | ☑ | ☑ |
>
> We group the prior work into 2 classes.
>
> 1. Several recent methods propose interpolating across multiple timepoints by locally optimizing between adjacent marginals. While these approaches are generally scalable, they do not recover the global coupling.
>     - SBIRR [1] is a simulation-free iterative scheme performing trajectory inference via pairwise bridge optimization. Nevertheless, this pairwise formulation fails to capture global trajectory consistency. As a result, SBIRR relies on informative priors to guide trajectory inference, making it sensitive to the choice of reference class and often impractical for real-world applications.
>     - MMFM [2] performs cubic spline-based interpolation using deterministic dynamics. While scalable, it relies on precomputed piecewise OT couplings and hence does not provide the joint optimal coupling. This impairs the temporal coherence of the trajectories and the accurate capture of the dynamics.
>     - MMSFM [3] proposes a stochastic flow matching that optimizes measured value splines in position space using overlapping windows of three consecutive marginals. While this improves upon MMFM's pairwise interpolation, the lack of global coupling and reliance on first-order dynamics hinder its ability to learn smooth, temporally coherent trajectories.
> 2. An alternative class of methods solves the mmSB without local approximations. However, these methods generally present low to moderate scalability capacity.
>     - DMSB [4] solves the mmSB in phase space using Bregman iterations and successfully retrieves the mmSB coupling. However, it suffers from scalability limitations due to the need to cache full SDE trajectories, leading to computational bottlenecks, error accumulation, and potential instability.
>     - Smooth SB [5] replaces Brownian priors with smooth Gaussian paths in phase space, achieving more temporally coherent and smooth trajectories. However, the belief propagation prohibits scaling in high dimensions.
>
> Our proposed algorithm directly addresses these limitations. The differentiating attributes of our 3MSBM compared to prior methods can be summarized as follows:
>
> - We lift the dynamics to phase space and minimize path acceleration, resulting in smoother low-curvature trajectories. This allows the model to capture trajectory trends typically lost in first-order systems—especially important when observations are sparse or irregular. As a result, the system can represent non-linear transitions and inflection points, enabling more realistic interpolation.
> - We solve the multi-marginal Brownian Bridge (3MBB; Theorem 3.1) objective analytically, and derive closed-form solutions for the conditional bridges across any number of $N$ marginals. This eliminates the need for costly numerical integrations of Eq. (5), which would scale poorly and accumulate approximation errors. By also avoiding approximations (e.g., local optimization of adjacent marginals and truncated policies), our methodology converges to the mmSB coupling across $N$ marginals.
> - Our matching framework allows us to learn the parameterized drift given a prescribed path in a simulation-free manner. Moreover, the iterative solutions of the learned drift remain close to the ground truth marginals throughout training, as opposed to prior methods that approximate the true marginals merely after convergence.
>
> ### **2.  Scalability of 3MBB**
>
> Our algorithm adopts a two-step architecture, each of which avoids costly numerical simulations.
>
> - Closed-form conditional bridge:
>
>     Notably, the analytic solution to the 3MBB objective allows us to compute the conditional bridge in Eq. (6) without requiring numerical integration or discretization, thereby eliminating both computational overhead and approximation errors. Consequently, this step remains independent of the number of marginals, as the conditional bridges are derived in closed form.
>
> - Bridge Matching:
>
>     The second step regresses the parameterized drift on the path prescribed in the previous step, in a simulation-free fashion, unlike more computationally expensive prior IPF-based methods.
>
>
> Next, we provide 4 additional experiments to validate our claims, which are detailed below.
>
> 1. We compare the total training time of our algorithm to DMSB across all experiments and observe that our 3MSBM is consistently and significantly faster in wall-clock time across all datasets.
>
>     | Time (hours:min) | EB-100 | GoM | LV | BAQ |
>     | --- | --- | --- | --- | --- |
>     | DMSB | 5:51 | 2:55 | 2:43 | 3:15 |
>     | 3MSBM | 2:18 | 0:34 | 0:21 | 2:03 |
>
> 2. We ablate the number of marginals in the EB-100 and GoM datasets, and report the training time per-epoch. We observe that the number of marginals has a negligible impact on training time per epoch, indicating that our algorithm scales well with increasing number of marginals.
>
>     - **Ablation on EB-100**
>     | Num of marginals | Time per epoch |
>     | --- | --- |
>     | 3 | 773 sec |
>     | 4 | 789 sec |
>     | 5 | 812 sec |
>
>     - **Ablation on GoM**
>     | Num of marginals | Time per epoch |
>     | --- | --- |
>     | 5 | 263 sec |
>     | 7 | 270 sec |
>     | 9 | 290 sec |
>
> 3. We present a breakdown of the time percentage attributed n each of the two steps of our algorithm: i) the iterative propagation of the conditional dynamics (*lines 4-9 in Alg. 1)* and ii) the Bridge Matching step *(line 10 in Alg. 1)* take in the EB-100 and GoM datasets. We observe that the computational complexity introduced by the first step of the algorithm, i.e., the iterative propagation of the conditional dynamics, remains negligible compared to the Bridge Matching step.
>
>
>     |  | Cond. Bridge | Bridge Matching |
>     | --- | --- | --- |
>     | EB-100 | 3.3 % | 96.7 % |
>     | GoM | 7.4 % | 92.6 % |
>
> 4. We ablate the batch size in the EB-100 dataset and report the peak memory usage compared to DMSB. We find that our method consistently requires less memory than the IPF-based DMSB, which encounters a memory error for larger batch sizes. This enables significantly larger batch sizes, even in high-dimensional settings, providing greater design flexibility without compromising performance.
>
>
>     | Batch Size | DMSB | 3MSBM |
>     | --- | --- | --- |
>     | 256 | 6.1 GB | 1.8 GB |
>     | 512 | 12.3 GB | 3.3 GB |
>     | 1024 | -- | 6.7 GB |
>     | 2048 | -- | 13.8 GB |
>
> ### **3. Robustness to un-smooth and multi-marginal data**
>
> We emphasize that a key feature of our method is the utilization of conditional bridges that explicitly condition on all future pinned marginals, as shown in Eq. (6). This design allows the learned dynamics to effectively “look ahead,” providing a global temporal context that enhances robustness to both irregular transitions and multi-modal marginals.
>
> Regarding robustness to less smooth data, our method outperformed state-of-the-art baselines on the Beijing Air Quality dataset (Section 4.3), comprising a time series with abrupt fluctuations and high volatility. To further validate the robustness of our method under multi-modal scenarios, we additionally consider:
>
> 1. A sequence of marginals starting from a univariate Gaussian $(t_0)$, transitioning to a 4-mode GMM $(t_1)$, followed by an 8-mode GMM $(t_2)$, and returning to a univariate Gaussian $(t_3)$.
>
>
>     |  | $\mathcal{W}_2$   @   $t_0$ | $\mathcal{W}_2$   @   $t_1$ | $\mathcal{W}_2$   @   $t_2$ | $\mathcal{W}_2$   @   $t_3$ |
>     | --- | --- | --- | --- | --- |
>     | DMSB | 0.05 | 0.06 | 0.4 | 0.09 |
>     | 3MSBM | 0.01 | 0.03 | 0.07 | 0.01 |
>
> 2. A more complex geometry starting from a 9-mode GMM arranged in a rhomboid shape $(t_0)$, transitioning to five Swiss rolls, with four placed at the corners of a rectangle and one in the center $(t_1)$.
>
>
>     |  | $\mathcal{W}_2$   @   $t_0$ | $\mathcal{W}_2$   @   $t_1$ |
>     | --- | --- | --- |
>     | DMSB | 0.07 | 0.14 |
>     | 3MSBM | 0.03 | 0.05 |
>
> In both settings, we compute the $\mathcal{W}_2$ distance between generated samples and ground truth marginals, comparing our method to DMSB. Our approach consistently yields lower $\mathcal{W}_2$ errors, demonstrating superior fidelity in capturing complex and multi-modal dynamics.
>
> ### **4. Typo Correction**
>
> We thank the reviewer for pointing out this typo. It has been corrected: $R$  refers to the cost matrices in the optimal control problem. In Algorithm 1, the number of iterations is denoted by 'Max_Iter' and the loop index by 'iter'.
>
> ---
>
> ### References
>
> 1. Multi-marginal Schrödinger Bridges with iterative reference refinement.
> 2. Modeling complex system dynamics with flow matching across time and conditions.
> 3. Multi-marginal stochastic flow matching for high-dimensional snapshot data at irregular time points.
> 4. Deep momentum multi-marginal Schrödinger Bridge
> 5. Trajectory inference with smooth Schrödinger Bridges

---

> > ### Comment · Reviewer_Pewn · 2025-08-04
> >
> > I feel all my concerns were substantially addressed. I will raise my score to 4.

---

### Official Review · Reviewer_mTyy · 2025-07-01

**Clarity:** 3
**Significance:** 4
**Originality:** 4
**Rating:** 5
**Confidence:** 4

**Summary:**

The paper proposes 3MSBM, a simulation-free matching algorithm for multi-marginal Schrödinger Bridge problems in phase space. By deriving recursive formulas for optimal acceleration, it achieves smooth, temporally coherent trajectories while preserving marginal constraints throughout training. Experiments across synthetic and real-world datasets show that 3MSBM outperforms prior methods in both accuracy and efficiency.

**Questions:**

----
* If the algorithm were implemented using only forward dynamics, any suboptimality in the initial Markov projection (i.e., the objective in Equation (9)) would propagate forward, accumulating bias and ultimately leading to a biased estimate of the MMSB solution. It appears that the authors do adopt a forward–backward scheme as described in [1], which mitigates this issue. However, to the best of the reviewer's understanding, this important detail is not clearly stated in the main paper. Since many potential readers—especially those working in population dynamics—may not be familiar with the Schrödinger Bridge Matching (SBM) framework, the reviewer recommends that the authors explicitly describe this forward–backward structure in the main paper, particularly in Section 3.3.
----

* For Theorem 3.5, it appears that the notation $\mathbb{P}$ refers to the path measure of the underlying stochastic process (i.e., the solution to the SDE). However, this notation is used without being explicitly introduced, and related concepts such as the path space are also not defined. Providing a brief clarification would improve readability, especially for readers less familiar with measure-theoretic formulations of stochastic processes.

----

* The 3MBB framework introduces the dynamic programming principle by simulating backward ODEs to compute the optimal control. However, when the number of marginals $N$ is large, this backward integration may accumulate numerical errors, leading to a biased estimate of the initial control $u_0$. This bias subsequently propagates through the forward dynamics, since the forward bridge is conditioned on a suboptimal initial segment. While the authors derive an analytic expression for the optimal control in Eq. (7) and Appendix B.2, the derivation is restricted to up to five marginals. A closed-form analytic solution for general N could eliminate such approximation errors and improve scalability. Given the structure shown in the appendix, is it possible to derive a general recurrence relation for the $N$-marginal case?
----

* As the reviewer understand, SBIRR refines a predefined prior process in both the Lotka–Volterra (LV) and Gulf of Mexico (GoM) experiments, where the chosen priors already provide a strong approximation of the underlying dynamics. However, in general settings, assuming access to such informative priors that already capture the system behavior is often unrealistic. In contrast, the proposed 3MSBM appears to model the dynamics effectively without relying on strong prior assumptions. Could the authors clarify whether 3MSBM also benefits from a similar modeling choice, or does it learn the dynamics purely from data?

----

* Prior approaches [2] rely on computationally expensive Bregman iterations to enforce multiple constraints, resulting in long training times. Compared to this, I wonder how much training time the proposed method actually saves. Since SBM methods are generally known to converge faster than IPF-type objectives, and DMSB is based on such an IPF framework, it would be helpful to clarify whether 3MSBM also provides a significant advantage in terms of training efficiency.

---

* (Minor comment) Equation (4) is missing the differential $\mathrm{d}t$ for the control.

----
    [1] Diffusion Schrödinger Bridge Matching.
    [2] Deep Momentum Multi-Marginal Schrödinger Bridge.

**Ethical Concerns:**

["NO or VERY MINOR ethics concerns only"]

**Final Justification:**

Momentum-SB methods capture underlying physics directly from data via their momentum term, providing a compelling framework for data-driven physical modeling. The proposed 3-MSBM preserves this strength while markedly reducing training time and memory consumption compared with earlier momentum-SB approaches. Experiments confirm its effectiveness and scalability. Reviewers pointed out that several key concepts were missing from the main text and had to be inferred from the figures; the authors have committed to clarifying these points in the revision.

**Limitations:**

yes

**Quality:**

4

**Strengths And Weaknesses:**

**Strengths**

* 3MSBM is a new matching framework that extending Schrödinger Bridges (SB) under multiple marginal constraints in phase space. This allows the model to generate smooth and temporally coherent trajectories that satisfy multiple positional constraints across time.

* This work proposes a matching-based algorithm by deriving recursive formulas for optimal acceleration tailored to multi-marginal settings. These formulas extend Brownian bridges to momentum systems and enable simulation-free training using analytic Gaussian paths, making it possible to generalize previous matching frameworks to the phase-space multi-marginal SB setting.

* Experiments on diverse datasets, demonstrate consistent improvement over state-of-the-art methods. 3MSBM achieves lower transport errors and better marginal fidelity in both low and high-dimensional regimes.

----

**Weaknesses**

See questions below.

---

> ### Author Rebuttal · Authors · 2025-07-31
>
> We truly appreciate the encouraging comments of the reviewer, as well as their useful remarks. In the following, we address the points raised by the reviewer.
>
> ### **1. Comment on Forward-Backward scheme**
>
> We thank the reviewer for their comment. This is true, our algorithm does involve a forward-backward scheme, which will be explicitly stated in Section 3.3 to avoid confusion.
>
> ### **2. Comment on formally defining path measure $\mathbb{P}$**
>
> We thank the reviewer for identifying this omission. In the revision, we will formally define the path measure $\mathbb{P}$ associated with the learned dynamics, and $\mathbb{P}^i$ as the learned dynamics at the $i^\text{th}$ iterate of our algorithm before Theorem 3.5.
>
> ### **3. Recursive formula for conditional acceleration**
>
> We thank the reviewer for this important and insightful comment. While we agree with the reviewer’s conjecture that naive backward simulation of the ODEs in Eq (5) to compute the optimal control can scale poorly as $N$ grows, we emphasize that this is not practically implemented by our 3MBB. Specifically, Eq. (5) was never simulated throughout 3MBB training. Instead, we are able to compute an analytic solution to Eq. (5) for arbitrary $N$, as proposed by the reviewer.
>
> The recursive expression for arbitrary $N$ is formalized in Prop 3.2, where the analytic formulation ensures that both the computational cost and approximation error remain independent of the number of marginals, effectively addressing the scalability and stability concerns associated with numerical solvers. Below, we present the general recursive formula, broken down in two steps, for obtaining the conditional bridge for an arbitrary $n^{\text{th}}$ segment with $n<N-1$, i.e., $n$ is not the last segment.
>
> 1. Expressions for the time-dependent functions $C_1^n(t), C_2^n(t), C_3^n(t)$
>
>     For convenience, let us define the following functions corresponding to the $n^\text{th}$ segment: $t\in[t_n, t_{n+1})$
>
>     $$ z_1^{ (n)}(t) = 3t - 3t_{n+1} - 3  \qquad z_2^{ (n)}(t) = 3t - 4 - 3t_{n+1} \qquad z_3^{ (n)}(t) = 6t - 6t_{n+1} - 3 \qquad z_4^{ (n)}(t) = 6t - 6t_{n+1} - 4 $$
>
>     $$ z_5^{ (n)}(t) = 4t - 4t_{n+1} - 3 \qquad z_6^{ (n)}(t) = 4t - 4 - 4t_{n+1} \qquad  z_7^{ (n)}(t) = 6t - 6t_{n+1} + 3 \qquad z_8^{ (n)}(t) = 6t - 6t_{n+1} + 4 $$
>
>     It follows that for the arbitrary $n^{\text{th}}$ segment, the corresponding time-dependent functions $C_1^n(t), C_2^n(t), C_3^n(t)$, are given by
>     $$C_1^{\small (n)}(t) = \frac{-3(\alpha^{(n)} z_3^{(n)}(t) + \beta^{(n)} z_4^{(n)}(t))}{(t-t_{n+1})^2*(\alpha^{(n)} z_1^{(n)}(t) + \beta^{(n)} z_2^{(n)}(t))} $$
>     $$C_2^{(n)}(t) = \frac{3(\alpha^{(n)} z_5^{(n)}(t) + \beta^{(n)} z_6^{(n)}(t))}{(t-t_{n+1})*(\alpha^{(n)} z_1^{(n)}(t) + \beta^{(n)} z_2^{(n)}(t))} $$
>     $$C_3^{(n)}(t) = \frac{6(\alpha^{(n)}+ \beta^{(n)})}{(\alpha^{(n)} z_1^{(n)}(t) + \beta^{(n)} z_2^{(n)}(t))}$$
>
>     where $\alpha^{(n)}, \beta^{(n)}$ are segment-specific coefficients that shape the conditional bridge for the given segment $n$. These are recursively computed using the coefficients of the subsequent segment $n+1$, as follows:
>     $$\alpha^{(n)} = \alpha^{(n+1)} z_1^{(n)}(t_{n+1}) + \beta^{(n+1)} z_2^{(n)}(t_{n+1}) $$
>     $$\beta^{(n)} = 4(\alpha^{(n+1)} +\beta^{(n+1)})$$
>
>     with the coefficients of the second-to-last segment $N-1$ given by: $\alpha^{(N-1)}=0, \ \beta^{(N-1)}=1$.
>
> 2. The $\lambda^{(n)}$ coefficients
>
>     Because of the recursive nature of the backward ODE of $r_t$, with the terminal conditions depending on all succeeding segments, we find that the coefficients $\lambda^{(n)}$ converge to the following exponential decay with an absolute error $<10^{-6}$ for segments more than 6 marginals before the last one:
>
>     $\lambda_j^{(n)} = (-1)^{j}1.608e^{-1.317(j-1)}$ ,   for     $j = 2, \dots, N$         and       $\lambda_1 = -\sum_{j=2}^N \lambda_j^{(n)}$
>
>     Conversely, if the number of future marginals is smaller than 5, the $\lambda^{(n)}$ are given by the values shown in the examples in Appendix B.2.1.
>
>
> We had initially presented the bridges for up to 5 segments for simplicity, but we will enrich Appendix B.2.1 with the discussion above to better describe the procedure and the steps needed to derive the conditional bridge for an arbitrary $n^\text{th}$ segment: $t\in[t_n, t_{n+1})$.
>
> ### **4. Discussion on SBIRR**
>
> We emphasize that, unlike SBIRR, which assumes prior knowledge of the underlying system behavior, our proposed 3MSBM learns the dynamics and achieves highly accurate trajectory inference purely from the data. This is made possible by two key components of our framework:
>
> 1. By incorporating the velocity into the state, our 3MSBM captures *momentum-like effects* that encode trends in the data—such as speeding up or slowing down—that are otherwise lost in first-order models. This is particularly important in settings with sparse or irregular observations, as it enables the model to infer smoother and more physically realistic trajectories through intermediate timepoints, including non-linear transitions and inflection points.
> 2. Our conditional bridge formulation solves for the joint optimal coupling across all marginals, which allows the dynamics at any given time to incorporate information from *all future pinned points*. This provides a form of implicit "look-ahead" guidance, helping the model infer trajectories that are globally consistent and faithful to the underlying physical or biological system.
>
> ### **5. Comparisons between 3MSBM and DMSB**
>
> Our 3MSBM as a matching framework provides enhanced scalability compared to prior IPF based methods solving the mmSB problem, such as Deep Momentum Multi-Marginal Schrödinger Bridge (DMSB;[1]). To empirically verify our claims:
>
> 1. We compare the total training time of our algorithm to DMSB on all datasets and observe that 3MSBM is consistently and significantly faster in wall-clock time across all datasets.
>
>
>     | Time (hours:min) | EB-100 | GoM | LV | BAQ |
>     | --- | --- | --- | --- | --- |
>     | DMSB | 5:51 | 2:55 | 2:43 | 3:15 |
>     | 3MSBM | 2:18 | 0:34 | 0:21 | 2:03 |
>
> 2. We ablate the batch size in the EB-100 dataset and report the peak memory usage compared to DMSB. We find that our method consistently requires less memory than the IPF-based DMSB, which encounters a memory error for larger batch sizes. This enables significantly larger batch sizes, even in high-dimensional settings, providing greater design flexibility without compromising performance.
>
>     | Batch Size | DMSB | 3MSBM |
>     | --- | --- | --- |
>     | 256 | 6.1 GB | 1.8 GB |
>     | 512 | 12.3 GB | 3.3 GB |
>     | 1024 | -- | 6.7 GB |
>     | 2048 | -- | 13.8 GB |
>
>
> ### **6. Typo Correction**
>
> We thank the reviewer for identifying this typo. It has been corrected in the revised manuscript.
>
> ---
>
> ### References
>
> 1.  Deep momentum multi-marginal Schrödinger Bridge

---

> > ### Comment · Reviewer_mTyy · 2025-08-04
> >
> > Thank you for the response. Momentum-SB methods stand out because their momentum-like effects let them capture the underlying physical system in a fully data-driven way. The proposed Matching algorithm preserves those advantages while cutting both training time and memory usage compared with earlier momentum-SB approaches. Assuming the revised manuscript now clearly positions these contributions as outlined in your response, I will maintain my recommendation to accept the paper.

---

### Official Review · Reviewer_dYDf · 2025-07-02

**Clarity:** 3
**Significance:** 4
**Originality:** 3
**Rating:** 4
**Confidence:** 3

**Summary:**

This paper introduces Momentum Multi-Marginal Schrödinger Bridge Matching (3MSBM), a novel framework for inferring smooth, temporally coherent stochastic trajectories that simultaneously satisfy multiple positional constraints. By lifting dynamics into phase space (position + velocity), the authors extend classical Schrödinger Bridge theory to handle an arbitrary sequence of intermediate distributions.

**Questions:**

1. There are notable inconsistencies between expressions (2) and (3), (4). Specifically, (2) involves the term $u_tdt$, while (4) only contains
$u_t$; in addition, (2) does not include 's.t.', and (4) lacks the definitions of $A$ and $g$.

2. Is there any condition imposed on $u_t$? In particular, is it required that the first half of $u_t$ be zero?

3. Essentially, is (2) a second-order equation? If so, why is it expressed in the current form?

4. The ODE version, i.e., dynamic optimal transport, corresponds exactly to optimal transport. Does the method proposed in the paper admit a similar interpretation in terms of optimal transport?

5. It is claimed that existing methods struggle to capture long-range temporal dependencies, which affects the coherence of the inferred trajectories. However, it remains unclear why formulating the problem using a second-order SDE addresses this specific limitation.

**Ethical Concerns:**

["NO or VERY MINOR ethics concerns only"]

**Final Justification:**

The response has successfully resolved all the issues I had.

**Limitations:**

Yes

**Quality:**

3

**Strengths And Weaknesses:**

Strengths
1. A simulation-free, matching-based method that preserves all intermediate marginals throughout training, avoiding trajectory caching and time-discretization issues.

2. Generalization of momentum Brownian bridges to the multi-marginal setting, with analytic expressions for the optimal control and conditional acceleration.

3. Provable stable convergence and efficient implementation that scale to high dimensions and many marginals.

4. Demonstrated superior performance over state-of-the-art multi-marginal methods (DMSB, SBIRR, Smooth SB, MMFM) on synthetic dynamics (Lotka–Volterra), real-world flows (Gulf of Mexico, Beijing air quality), and high-dimensional single-cell differentiation data.

Weaknesses

Several parts of the manuscript suffer from unclear wording and a high density of technical or grammatical errors, which hinder readability and understanding.

---

> ### Author Rebuttal · Authors · 2025-07-31
>
> We would like to thank the reviewer for their positive comments, as well as their constructive criticism and further suggestions for improving our work. In the following, we address the points raised by the reviewer.
>
>
> ### **1. Typo Corrections**
>
> We appreciate the reviewer’s suggestions to improve the clarity and readability of our work. We have carefully reviewed the paper and made several corrections that will be incorporated into the revised version, as we can not update the manuscript during the rebuttal phase. More specifically, we have extensively addressed and corrected technical typos in Equations, such as:
>
> - Eq. (4) was missing a differential $dt$, hence the corrected $dm_t = Am_t dt+udt + \mathbf{g}dW_t$.
> - Eq. (2) was corrected to include the ‘s.t.’ to explicitly denote the constraints of our optimization problem in Eq. (2).
> - The definitions of $A$ and $\mathbf{g}$ in Eq. (4) follow from Eq. (2). However, we included them in Eq. (4) as well to improve readability.
>
> Additionally, we thoroughly reviewed the manuscript and corrected grammatical errors. These include typographical errors (“phase state” → “phase space”), punctuation/parallel‑structure fixes, article misuse and noun phrase issues (e.g., “globally optimality coupling” → “globally optimal coupling”), and bracket mismatches in Algorithm 1.
>
> We are confident these improvements have greatly enhanced readability and have addressed the clarity concerns raised.
>
> ### **2. Notation clarifications**
>
> Our current notation follows prior multi-marginal and momentum works [1,3]. Our system is indeed second-order, consisting of the two following differential equations
>
> $$dx_t = v_tdt $$
> $$dv_t = a_tdt + \sigma dW_t$$
>
> where $x_t$  is the state, $v_t$  is the velocity, $a_t$  is the acceleration, $W_t$  is the Brownian motion and $\sigma$  is a diffusion coefficient for the noise being injected only through the velocity channel.  Thus, the two differential equations can be rewritten in a more compact form as
>
> $$
> \begin{bmatrix}dx_t\\\ dv_t\end{bmatrix} = \bigg(\begin{bmatrix}0&1\\\0&0\end{bmatrix}\begin{bmatrix}x_t\\\ v_t\end{bmatrix} + \begin{bmatrix}0\\\ a_t\end{bmatrix} \bigg)dt + \begin{bmatrix}0&0\\\0&\sigma\end{bmatrix}dW_t
> $$
> By defining the augmented state $m_t =\begin{bmatrix}x_t\\\ v_t\end{bmatrix}$, $A=\begin{bmatrix}0&1\\\0&0\end{bmatrix}$, $u_t = \begin{bmatrix}0\\\a_t\end{bmatrix}$, and $\mathbf{g} = \begin{bmatrix}0&0\\\0&\sigma\end{bmatrix}$, we can write the augmented SDE above in a more concrete fashion, as follows:
>
> $$
> dm_t = (Am_t + u_t)dt + \mathbf{g}dW_t
> $$
>
> Therefore, the control $u_t$  is not constrained in any manner, rather than it is by definition that the control is injected only through the velocity channel through the acceleration, and hence follows the aforementioned formulation.
>
> To avoid confusion, we have revised Eq. (2) and (3) & (4) to enhance readability. In particular, we now explicitly include the differential equations as part of the constraints and have restructured the optimization objectives for better presentation. More explicitly, Eq. (2) is now expressed as:
> $$
> \min_{a_t} \int_0^1 \mathbb{E_\text{$p_t$}} \Bigr[\frac{1}{2}||a_t||^2\Bigr ] dt,
> $$
> $$
> \text{s.t.}\quad dx_t =v_t dt, \\quad v_t= a_tdt+\sigma dW_t, \\quad x_n\sim q_{n} = \int \pi_n(x, v) dv_n, \\quad n= \\{ 0, 1, \dots, N \\}
> $$
> and Eqs. (3) and (4) are rewritten as follows:
> $$
> \min_{a_t} \int_0^1  \frac{1}{2}||a_t||^2 dt + \sum_{n=1}^N (m_{n} - \bar{m}_{n})^\intercal R (m_n - \bar m_n)
> $$
> $$
> \text{s.t.}\quad  dx_t =v_t dt , \quad dv_t= a_tdt+\sigma dW_t , \quad m_0 = \bar{m}_0 .
> $$
>
> ### **3. Optimal Transport Interpretation**
>
> In this work, we focus on stochastic dynamics to accurately model the inherent randomness present in many real-world systems, such as those in meteorology and biology. The multi-marginal Schrödinger Bridge (mmSB) problem [2] has been shown to be the stochastic counterpart of optimizing measured value splines in the Wasserstein space [4], which can be interpreted as the multi-marginal Optimal Transport (mmOT) problem [5]. Notably, our 3MSBM provably converges to the mmSB solution — i.e., the joint optimal coupling over the set of $N$  marginals. Therefore, our method provably solves the stochastic equivalent of the momentum formulation of the mmOT problem.
>
> ### **4. Motivation for Second-Order Dynamics**
>
> To capture coherent long-term dependencies in sparse and irregularly sampled time series, we incorporate second-order dynamics into our dynamical systems and optimize them to minimize the acceleration of the path. This results in minimum curvature paths leading to smoother and consistent trajectories. Crucially, the inclusion of velocity into the dynamics provides information to the model about past trends—through the acceleration—that is otherwise lost in first-order systems. In contrast, first-order systems are characterized by memoryless dynamics, which lack the capacity to preserve momentum trends, often resulting in interpolated trajectories with unnatural *kinks,* as illustrated in Figure 1 of the manuscript, or discontinuities—phenomena that are statistically and physically implausible. To mitigate, prior methods based on first-order models typically rely on highly informative priors that encode the system’s behavior. However, such priors are rarely available in real-world scenarios, making this assumption impractical.
>
> By accounting for momentum, the model can represent non-linear transitions and inflection points in the underlying dynamics, enabling more realistic interpolation. This is especially critical when observations are limited to coarse or widely spaced timepoints, as it allows the system to preserve natural tendencies like speeding up or slowing down. Moreover, second-order dynamics allow us to more faithfully capture the behavior of many natural physical and biological systems, which inherently follow second-order laws—such as Newtonian or Hamiltonian dynamics—thereby making our model both more physically plausible and better aligned with real-world systems.
>
> In our proposed methodology, leveraging second-order dynamics allows us to infer physically plausible trajectories across arbitrary numbers of time marginals. By jointly optimizing across all timepoints, our method retrieves globally optimal couplings and captures the dynamics in a statistically and physically consistent manner. This results in long-range temporal inference that is temporally coherent and accurately represents the underlying physics of the system. Our experiments in Section 5 demonstrate that this leads to significantly improved performance over state-of-the-art methods, both qualitatively and quantitatively. In the revision, we have modified the final two paragraphs of the introduction to include the benefits of the formulation of the problem using a second-order SDE as discussed above.
>
> ---
>
> ### References
>
> 1. Deep momentum multi-marginal Schrödinger Bridge
> 2. Stochastic bridges of linear systems
> 3. Multi-marginal Schrödinger Bridges
> 4. A smoothest curve approximation
> 5. Measure-valued spline curves: an optimal transport viewpoint

---

> > ### Comment · Reviewer_dYDf · 2025-08-04
> >
> > I truly appreciate the comprehensive reply and the detailed explanation. It has resolved all the questions I had.

---

### Note · Authors · 2025-08-13

We thank the reviewers for their constructive feedback and for recognizing our theoretical contribution (dYDf, mTyy, wHqP), extensive experimentation (dYDf, mTyy, Pewn), and the clarity of our paper (Pewn, wHqP). In the rebuttal, we thoroughly addressed all reviewer concerns by clarifying key aspects of our method and adding new experiments, with all reviewers explicitly confirming that their questions and issues were resolved.

We have committed to incorporating all clarifications and additional experiments in the revision, including:
1. Relation to Optimal Transport (dYDf): We clarified that 3MSBM solves the stochastic variant of the momentum multi-marginal OT, with dynamics reflecting the inherent randomness of real-world systems.
2. Advantages of second-order dynamics (dYDf, Pewn, wHqP): We explained how lifting dynamics to phase space allows better modeling of coherent long-term dependencies, a critical feature for multi-marginal problems.
3. 3MSBM vs prior methods (Pewn): We highlighted how our method uniquely combines: (i) scalability, (ii) globally optimal coupling, and (iii) smooth, low-curvature trajectories, contrasting with prior methods that either sacrifice optimality for scalability (SBIRR, MMFM, MMSFM) or vice versa (DMSB, Smooth SB).
4. Details on conditional bridges (wHqP, mTyy): We clarified the exact solution to the linear conditional acceleration and general formula for any number of marginals *without* requiring numerical integration, ensuring scalability and stability.
5. Typo fixes (dYDf, mTyy): We corrected grammatical issues, typos, and clarified our notation.
6. Experiments (Pewn, wHqP, mTyy): We conducted additional experiments to empirically verify the efficacy, robustness, and scalability of 3MSBM:
   - Expanded interpolation results across all datasets, with more baselines and a unified metric, showing 3MSBM consistently outperforms all baselines.
   - Compared 3MSBM's scalability against DMSB via training-time analysis and batch size ablation, showing 3MSBM trains significantly faster while using less memory.
   - Ablated the number of marginals, showing 3MSBM scales well as their number increases, since training time is negligibly impacted.
   - Evaluated on multimodal marginals showcasing the robustness of 3MSBM in learning complex dynamics.

With its clarified theoretical novelty and demonstrated scalability, we are confident that 3MSBM marks a significant advancement for learning complex dynamics from multiple snapshots.

---

### Decision · Program_Chairs · 2025-09-17

**Decision:**

Accept (poster)

**Comment:**

This paper introduces momentum multi marginal Schrödinger bridge matching, a method for inferring smooth and continuous-time trajectories of stochastic systems from multiple, sparse snapshots. The core innovation is to address this as a multi marginal problem in phase space. This method learns a single, globally coherent trajectory rather than relying on pairwise interpolation.

Reviewers unanimously vote to accept this work after rebuttal. The strong rebuttal by the authors addressed all initial reviewer concerns and led two reveiwers to increase their score. The main strengths of this work are its theoretical soundness identifying and solving a key gap in the existing literature. Paired with strong empirical evaluation.

Initial weaknesses were noted around clarity and initial experiments, both fully resolved during the rebuttal. For these reasons I recommend acceptance at this time.